# Advances and Perspectives in the Management of Varicella-Zoster Virus Infections

**DOI:** 10.3390/molecules26041132

**Published:** 2021-02-20

**Authors:** Graciela Andrei, Robert Snoeck

**Affiliations:** Laboratory of Virology and Chemotherapy, Rega Institute for Medical Research, KU Leuven, 3000 Leuven, Belgium; robert.snoeck@kuleuven.be

**Keywords:** varicella-zoster virus (VZV), chickenpox, HZ, shingles, nucleoside analogues, helicase-primase inhibitors, valnivudine hydrochloride (FV-100), valomaciclovir stearate, amenamevir, anti-VZV drugs

## Abstract

Varicella-zoster virus (VZV), a common and ubiquitous human-restricted pathogen, causes a primary infection (varicella or chickenpox) followed by establishment of latency in sensory ganglia. The virus can reactivate, causing herpes zoster (HZ, shingles) and leading to significant morbidity but rarely mortality, although in immunocompromised hosts, VZV can cause severe disseminated and occasionally fatal disease. We discuss VZV diseases and the decrease in their incidence due to the introduction of live-attenuated vaccines to prevent varicella or HZ. We also focus on acyclovir, valacyclovir, and famciclovir (FDA approved drugs to treat VZV infections), brivudine (used in some European countries) and amenamevir (a helicase-primase inhibitor, approved in Japan) that augur the beginning of a new era of anti-VZV therapy. Valnivudine hydrochloride (FV-100) and valomaciclovir stearate (in advanced stage of development) and several new molecules potentially good as anti-VZV candidates described during the last year are examined. We reflect on the role of antiviral agents in the treatment of VZV-associated diseases, as a large percentage of the at-risk population is not immunized, and on the limitations of currently FDA-approved anti-VZV drugs. Their low efficacy in controlling HZ pain and post-herpetic neuralgia development, and the need of multiple dosing regimens requiring daily dose adaptation for patients with renal failure urges the development of novel anti-VZV drugs.

## 1. Introduction

Varicella-zoster virus (VZV), also known as human herpesvirus 3 (HHV-3) has a double-stranded DNA genome of 125 kb, encoding for approximately 71 open reading frames (ORFs). The herpesvirus family includes three subfamilies, α, β, and γ-herpesvirinae. VZV together with herpes simplex virus 1 and 2 (HSV-1 and HSV-2) belong to the α-herpesvirinae, characterized by establishment of latency in neurons.

VZV is the causative agent of chickenpox (varicella), a common infantile illness. Like all herpesviruses, VZV undergoes a lifelong latent state following primary infection. During latency, the viral DNA persists in the dorsal root ganglia and cranial root ganglia. VZV reactivation produces skin lesions characteristic of herpes zoster (shingles), causing significant morbidity, but rarely mortality [1]. Herpes zoster (HZ) is characterized by a localized rash in a unilateral, dermatomal distribution and is often associated with severe neuropathic pain.

VZV is highly infectious and enters the body via the respiratory tract, followed by rapid spread from the pharyngeal lymphoid tissue to circulating T lymphocytes [2]. After 10–21 days, the virus arrives at the skin, producing the typical vesicular rash characteristic of varicella. In most individuals, VZV infection results in lifelong immunity. During VZV reactivation, the virus is transported along microtubules within sensory axons to infect epithelial cells, usually without viremia. This gives a skin rash within the dermatome innervated by a single sensory nerve [the trigeminal (cranial nerve), cervical and thoracic sensory nerves are the most common nerves involved in VZV reactivation]. The virus can also be transmitted by fomites from varicella and shingles skin lesions. Reactivation from latency occurs when there is a weakening in cell-mediated immunity (CMI) as a natural consequence of aging since VZV-specific T cells lose their ability to proliferate with age or with immune suppression. Risks factors for HZ include older age, CMI dysfunction, diabetes, female gender, genetic susceptibility, physical trauma, recent psychological stress, and white race [2].

The narrow host range (VZV infection is highly restricted to humans) coupled with the highly cell-associated nature of the virus, has hampered the development of a reliable animal model that mimics VZV diseases in human, thereby hindering VZV pathogenesis studies.

## 2. Clinical Characteristics of VZV Infections

Primary VZV infection generally occurs in childhood and is usually mild but complications can occur in adults and in immunocompromised patients. Disseminated varicella infections with multi-organ failure have been reported in both immunocompetent [3,4,5] and immunocompromised patients [5,6,7]. Although VZV reactivation leading to HZ may occur at any age, the highest incidence is seen in the elderly and among immunocompromised individuals (HIV/AIDS individuals or patients suffering from malignancies, organ or hematopoietic stem cell transplant (HSCT) recipients or persons receiving high-dose corticosteroid therapy). Age-related impaired CMI facilitates VZV reactivation from latency. HZ affects up to 25% of human beings during their lifetime, with 50% of persons being aged 80 years or more [8]. HZ is rarely life threatening but it is associated with a number of acute syndromes, including a vesicular rash and pain. It can lead to prolonged pain, known as post-herpetic neuralgia (PHN), a debilitating condition, which can be very difficult to manage, mainly in the elderly where the disease tends to be more serious [9,10]. PHN is associated with a loss of physical function, encompassing fatigue, anorexia, weight loss, reduced mobility, physical inactivity, sleep disturbance (especially insomnia) resulting in a loss of social contact. A positive correlation between enhanced severity of pain and higher interference with daily activities was reported [9]. Ophthalmological manifestations of VZV may also occur. About 10–15% of HZ cases are subtyped as HZ ophthalmicus (HZO), occurring when the virus is reactivated in the ophthalmic division of the trigeminal nerve [11]. Approximately 50% of individuals with HZO develop ophthalmic complications [12]. Chronic ocular inflammation, loss of vision, and debilitating pain can be permanent sequelae of HZO.

Similar to other α-herpesviruses, VZV is able to infect the central nervous system (CNS) to cause encephalitis, either as a complication of varicella or HZ, but also in the absence of rash. About 50% of VZV encephalitis cases occur in individuals with immunosuppressive conditions, in whom viral reactivation is more common. A high viral diversity and mixed infections with different clades in the cerebrospinal fluid (CSF) from patients with VZV encephalitis, which can be explained by viral reactivation from multiple neurons, may contribute to the pathogenesis of VZV encephalitis [13].

VZV infections among immunocompromised patients can be more severe, of longer duration than in immunocompetent individuals and may have unusual clinical presentation with multidermal involvement and hyperkeratotic skin lesions [14,15]. VZV can be associated with severe acute retinal necrosis (ARN), a disease with poor prognosis; typically occurring in immunocompetent individuals though it can also occur in immunocompromised patients [16,17]. Unlike VZV necrotizing retinitis, progressive outer retinal necrosis (PORN), a form of VZV chorioretinitis, is found mostly in severely compromised people though, exceptionally, some cases of PORN have been described in immunocompetent persons [18,19,20,21]. Treatment strategies for PORN are rather unsuccessful and the disease can evolve fast leading to blindness within a few days or weeks.

## 3. Vaccination Strategies and Post-Exposure Prophylaxis

Varivax^®^ (the live-attenuated vaccine for varicella from Merck Sharp & Dohme Corp., Table 1), which is based on the VZV attenuated Oka strain (vaccine Oka, vOka), was licensed in United States in 1995 for children aged 12 to 18 months, leading to a substantial drop in the incidence of varicella. GlaxoSmithKline also developed and launched a live vOka preparation (Varilrix^®^), for immunization against VZV infections in 1998. To prevent mild breakthrough infections among vaccinated children following a single dose of vOka, a recommendation for a 2-dose schedule was given in 2006 to improve immunity to VZV. The live-attenuated heterogeneous vaccine preparation vOka is used routinely in many countries worldwide, which resulted in a significant decline in the incidence of varicella disease, with a decrease risk of developing zoster [22]. The varicella vaccine has a very good safety profile. Only less than 5% vaccine recipients can develop a papular or vesicular rash, usually at the site of infection within 6 weeks following vaccination. Vaccinated individuals have minimum risk for transmitting vOka to contact persons and transmission occurs only in the presence of a rash [23]. Generalized varicella-like rashes can develop rarely within 14 days after vaccination and are due to wild-type VZV incidentally acquired soon after vaccination. vOka also establishes latent infection in vaccine recipients and may very rarely reactivate to cause HZ. Vaccine-related HZ occurs less frequently and is less severe than wild-type VZV reactivation and always manifests as HZ of one dermatome. Like most live attenuated viral vaccines, the emergence of vaccine-wild type recombinant strains can occur. VZV frequently undergoes genetic recombination, and the vaccine strains have already been found in recombination events with the wild-type [24].

Two vaccines (Zostavax^®^ and Shingrix^®^, Table 1) are currently available to boost the CMI to VZV. Both vaccines proved to be safe and immunogenic and to reduce the incidence of HZ and PHN. The efficacy of the life-attenuated HZ vaccine (Zostavax^®^) decreases with age at the time of vaccination and with time since vaccination. The vaccine efficacy of recombinant HZ vaccine (Shingrix^®^) remains higher and appears to decline more slowly than that of the life-attenuated vaccine across all age groups. Both vaccines are cost-effective in individuals ≥50 years of age compared with no vaccination, especially among those 65–79 years of age and Shingrix appears to be more cost-effective than Zostavax^®^. Importantly, a live attenuated vaccine, such as Zostavax^®^, cannot be given to immunosuppressed patients and this patient population is at high risk for developing HZ. In contrast, Shingrix can be given to persons with impaired immune conditions [25].

VariZIG^®^ (Saol Therapeutics) is a VZV immune globulin preparation available for post-exposure prophylaxis of varicella in persons at high-risk for severe disease who lack immunity to VZV and who are ineligible for varicella vaccine. High-risk groups encompass immunocompromised persons (children and adults), newborns of mothers who developed varicella just before or shortly after delivery, premature babies, children younger than one year of age, adults without evidence of immunity and pregnant women. The time during which a patient may receive VariZIG^®^ after VZV exposure has been prolonged from 4 days to 10 days.

## 4. Treatment of VZV-Associated Diseases

Safe and effective anti-VZV therapy considerably contributed to diminish the morbidity and mortality associated with varicella and HZ, in particular in immunocompromised populations. The drugs licensed for the treatment of VZV-associated disease in United States include acyclovir (ACV), its oral prodrug valacyclovir (VACV), and famciclovir (FCV), the oral prodrug of penciclovir (PCV) (Table 2). 

Acyclovir [ACV, 9-(2-hydroxyethoxymethyl)guanine, Zovirax^®^], a structural analogue of the natural compound 2′-deoxyguanosine (Table 2), is a potent and selective inhibitor of VZV, HSV-1, HSV-2, and Epstein-Barr virus (EBV) with modest activity against human cytomegalovirus (HCMV). Acyclovir and its prodrug valacyclovir (L-valyl ester of acyclovir) are the gold standard for prophylaxis and therapy of HSV and VZV associated diseases. Acyclovir is converted to its monophosphate (ACV-MP) by the viral thymidine kinase (TK), which is further phosphorylated by cellular kinases to ACV-triphosphate (ACV-TP) (Figure 1). ACV-TP, the active form of acyclovir, is a competitive inhibitor with respect to the natural substrate dGTP (deoxyguanosine triphosphate) and it can be a substrate for the viral DNA polymerase and then be incorporated to the 3′-end of a synthesized DNA molecule. The DNA polymerase-associated 3′-5′exonuclease cannot excise the 3′-terminal ACV-TP residues, resulting in prevention of chain elongation as ACV lacks the 3′hydroxyl group required for DNA elongation. Acyclovir has limited oral bioavailability (15–30%) and restricted solubility in water (~0.2%, 25 °C), requiring relatively large doses and frequent administration to maintain plasma levels of acyclovir high enough to achieve viral inhibition. The valine ester of acyclovir, valacyclovir (VACV, Valtrex, Zelitrex) (Table 2) is a safe and efficacious prodrug (54% oral bioavailability). It is rapidly metabolized to yield acyclovir and the essential amino acid l-valine [26] due to a carrier-mediated intestinal absorption, via the human intestinal peptide transporter hPEPT1, followed by rapid conversion to acyclovir by ester hydrolysis in the small intestine. Several clinical studies demonstrated that valacyclovir has a safety profile comparable to that of acyclovir in HZ patients. Valacyclovir became a better option in the treatment of VZV infections since it requires a less frequent dosing regimen than acyclovir, contributing to increased patient adherence to therapy.

Penciclovir [PCV, 9-(4-hydroxy-3-hydroxymethyl-but-1-yl)guanine, Denavir^®^, Vectavir^®^], a 2′-deoxyguanosine analog, resembles acyclovir in chemical structure, mechanism of action and spectrum of antiviral activity (Table 2 and Figure 1) [27,28]. PCV-TP inhibits viral DNA polymerases through competition with 2′-deoxyguanosine triphosphate and incorporation into the synthesized viral DNA. Unlike ACV-TP, PCV-TP has two hydroxyl groups on the acyclic chain and thus PCV-TP is not an obligate chain terminator and can be incorporated into the extended DNA chain. Penciclovir is very poorly absorbed when given orally and thus, famciclovir (FAM), the diacetylester of 6-deoxypenciclovir, was developed as the oral prodrug, which has an oral bioavailability of 77%. Famciclovir is rapidly and extensively absorbed and efficiently converted to penciclovir in two steps: (1) removal of the two acetyl groups (the first one by esterases in the intestinal wall and the second one on the liver), and (2) oxidation at the six position catalyzed by aldehyde oxidase that accounts for the conversion of 6-deoxypenciclovir to penciclovir. Famciclovir is well tolerated in patients and is effective against HSV-1 and HSV-2 (for both therapy and long-term suppression of recurrent infections) and against VZV (for treatment of HZ).

Brivudine (BVDU), (*E*)-5-(2-bromovinyl)-2′-deoxyuridine, bromovinyldeoxyuridine Zostex^®^, Zonavir^®^, Zerpex^®^ (Table 2) is licensed for the therapy of HZ in several countries in Europe. This thymidine analogue is a highly selective antiviral agent active against HSV-1 and VZV [29]. Several congeners of brivudine have been synthesized, including BVaraU (sorivudine), the arabinofuranosyl counterpart of brivudine. The selective anti-HSV-1 and anti-VZV activity of brivudine is dependent on a specific phosphorylation of the compound by the HSV-1 and VZV TK to its monophosphate (BVDU-MP) and diphosphate (BVDU-DP) (Figure 2). Following conversion to the triphosphate form (BVDU-TP) by a nucleoside diphosphate (NDP) kinase, BVDU-TP competes with the natural substrate dTTP (deoxythymidine triphosphate) for the viral DNA polymerase, inhibiting the incorporation of dTTP into the viral DNA or, as an alternate substrate is incorporated leading to the formation of a structurally and functionally disabled viral DNA [29]. Approximately 90% is absorbed following oral administration of brivudine and about 70% of the oral dose is rapidly transformed to bromovinyluracil (BVU) during the first passage through the liver [30]. Brivudine is effective in the treatment of HZ, both the short-term (formation of new lesions) and long-term (prevention of PHN) effects, and is as efficient and/or convenient as the other anti-VZV drugs acyclovir, valacyclovir and famciclovir. A recent retrospective study compared efficiencies of valacyclovir, famciclovir and brivudine in terms of pain relief in HZ patients [31]. All three drugs were effective in treating pain in acute HZ with no significant difference between mild and moderate HZ patients. A significant reduction in intensity of pain was observed in several cases on day 3 (brivudine group), on day 7 (famciclovir group), and at 2–3 weeks (valacyclovir group). Significant side effects were not observed in any of the groups. Based on the results of this study, brivudine could be considered as the first choice to treat severe HZ cases considering that is administered once daily and can control pain earlier.

One important limitation for the use of brivudine is the absolute contraindication of the combination of brivudine with 5-fluorouracil or its oral prodrug capecitabine since BVU, the degradation product of brivudine, is a potent inhibitor of dihydropyrimidine dehydrogenase, the enzyme responsible for the first step in the catabolic pathway of pyrimidines (Figure 2). Dihydropyrimidine dehydrogenase is needed for 5-fluorouracil degradation and, thereby concomitant administration of 5-fluorouracil together with brivudine results in increased exposure to 5-fluorouracil (since BVU hampers 5-fluorouracil degradation to the inactive 5-fluoro-5,6-dihydrouracil product, significantly increasing 5-fluorouracil half-life) [32]. Clinicians should thus be aware of the life-threatening and possible fatal drug-drug interaction of brivudine and capecitabine/5-fluorouracil [33,34,35]. Sorivudine, like brivudine, is metabolized to BVU and therefore, its administration with 5-fluorouracil is contraindicated. Sorivudine, licensed in Japan in 1993 for the treatment of HZ, was withdrawn following several deaths related to the co-administration with 5-fluorouracil [36,37].

The helicase-primase inhibitor (HPIs) amenamevir {ASP2151, *N*-(2,6-dimethylphenyl)-*N*-[2-[4-(1,2,4-oxadiazol-3-yl)anilino]-2-oxoethyl]-1,1-dioxothiane-4-carboxamide} (Table 2) was approved for treatment of HZ in Japan in September 2017 while the phase I trial with this drug was halted in United States due to safety concerns. The herpesvirus helicase–primase complex (encompassing the helicase, the primase and a cofactor protein with 1:1:1 stoichiometry) possesses multiple enzymatic activities including DNA helicase, single-stranded DNA (ssDNA)-dependent ATPase and primase, all essential for viral DNA replication and viral growth. Agents targeting the helicase–primase complex represent a breakthrough in the development of anti-herpesvirus agents. The helicase–primase complex is well conserved among members of the herpesvirus family and genes encoding the VZV helicase subunit (ORF55), primase subunit (ORF6) and cofactor subunit (ORF52) share homology with, respectively, the HSV-1 UL5, UL52, and UL8 genes and with the HCMV genes UL105, UL70 and UL102.

Amenamevir is an oxadiazole phenyl derivative with potent activity against both VZV and HSV [38], while the two other classes of HPIs, i.e., the thiazole urea derivative Pritelivir (AIC316, BAY 57-1293) and the 2-aminothiazolylphenyl type, BILS 179 BS, have a limited antiviral spectrum inhibiting only HSV-1 and HSV-2. HPIs are virus-specific, have low in vitro toxicity, inhibit both reference and clinical viral strains, and are orally bioavailable and effective in different mouse models of HSV infection. They have a completely different mechanism of action (direct inhibition of the helicase-primase complex) compared to the classical anti-herpesvirus agents (target the DNA polymerase) and do not need activation by the viral TK. Thus, HPIs are active against viral mutants with a defective TK (TK−). Furthermore, combination of HPI with nucleoside analogues showed a synergistic effect in vitro, pointing at combination therapy as a potential approach for treating severe conditions, such as encephalitis or infections in patients with immunosuppression [39,40]. To confirm that the anti-VZV activity of amenamevir was due to inhibition of the VZV helicase-primase complex, an amenamevir VZV mutant was selected and characterized [41]. Sequencing analysis of ORF55 (helicase gene) and ORF6 (primase gene) of this mutant indicated three amino acid changes from the parent strain: N336K in the helicase motif IV, one of the six well-conserved sequence motifs in ORF55, R446H in ORF55 and N939D in ORF6. Notably, this mutant showed a marked defect in viral replication. Astellas Pharma originally developed amenamevir for treatment of genital herpes and HZ. The efficacy of amenamevir 2× daily was comparable to acyclovir twice a day (BID) for 3 days using the primary endpoint ‘time to lesion healing’ in a phase II dose-finding study in patients with genital herpes (NCT00486200) [42]. Time to healing was shorten by 1–2 days in both treatment groups compared to the placebo group. The results of the clinical trial (NCT00487682), sponsored by Astellas Pharma, to investigate the efficacy and safety of three different doses of ASP2151 compared to valacyclovir in subjects with HZ and to determine the recommended clinical dose, have not yet been reported. Following a phase 1, randomized, double blind, multiple dose, multicenter study (NCT00870441) to compare the safety of amenamevir to valacyclovir and placebo in healthy male and female subject, Astellas Pharma halted the clinical development of amenamevir due to treatment-emergent serious adverse events. Maruho Co., Ltd. (Kyoto, Japan) resumed the development of amenamevir and conducted a randomized, double-blind, valacyclovir-controlled phase 3 study to evaluate the efficacy and safety of amenamevir in Japanese HZ patients that receive the drug within 72 h following the start of the rash. The study included 751 patients who were randomly distributed to be treated with either amenamevir 400 mg or 200 mg per os, 1× per day or valacyclovir 1000 mg 3× per day (3000 mg total daily dose) for 7 days (NCT01959841) [43]. The cessation proportion of development of new lesions at day 4, i.e., day 4 cessation proportion, was considered the primary efficacy end-point and these proportions were 81.1% (197/243), 69.6% (172/247) and 75.1% (184/245), respectively, for amenamevir 400 and 200 mg and valacyclovir, with non-inferiority of amenamevir 400 mg to valacyclovir confirmed by a closed testing procedure. Secondary end-points (days to cessation of new lesions formation, complete crusting, healing, pain resolution and virus disappearance) were not significantly different among the three treatment groups. Amenamevir (400 and 200 mg) and valacyclovir 3000 mg were well tolerated and the proportions of patients experiencing drug-related adverse events were 10.0% (25/249), 10.7% (27/252) and 12.0% (30/249), respectively. This study showed that amenamevir 400 mg is effective and well tolerated for treatment of HZ in immunocompetent Japanese patients, leading to amenamevir approval for this indication in Japan. 

Cidofovir, the first FDA-approved acyclic nucleoside phosphonate (ANP) for intravenous treatment of HCMV retinitis in AIDS patients in 1996, is mostly used off-label for the intravenous or topical treatment of severe infections caused by various DNA viruses, including various herpesviruses other than HCMV, polyoma-, adeno-, pox- (molluscum contagiosum virus and orf virus) and human papillomaviruses (HPV). In the case of VZV, the drug is used off label for therapy of acyclovir and/or foscarnet resistant infections. Cidofovir has a phosphonate group bypassing the first phosphorylation step by viral kinases. Cellular kinases convert the drug to the active diphosphate form (CDVpp), which acts as a competitive inhibitor of the viral DNA polymerase, causing slow down elongation and premature chain termination during viral DNA synthesis (Figure 1). CDVpp inhibits viral DNA polymerases more potently than cellular DNA polymerases. The metabolites of the ANPs show an unusually long intracellular half-life, accounting for a long-lasting antiviral activity. The formation of CDVp-choline adduct serves as an intracellular reservoir for the mono-(CDVp) and diphosphonate (CDVpp), explaining the long-lasting activity of the drug. CDV has two important drawbacks that restrict its use in clinic. Due to its low oral bioavailability (<5%), the drug needs to be administered intravenously, normally once a week because of its long-lasting activity. CDV is also known for its dose dependent nephrotoxicity, which can be reduced by pre-hydration with at least 1 L of 0.9% saline solution intravenously before each CDV infusion and concomitant oral administration of probenecid.

The pyrophosphate analogue foscarnet (PFA, Foscavir^®^), a direct inhibitor of viral DNA polymerases, is independent of activation by the viral TK (Figure 1 and Table 2). Hence, PFA is the therapy of choice for acyclovir-resistant (ACV-R) VZV infections due to mutations in the viral TK gene,

## 5. Medical Need for New Antiviral Agents to Manage VZV-Associated Diseases

### 5.1. Management of PHN and other Complications

Current antiviral drugs available for HZ treatment significantly decrease the incidence of new lesion formation, accelerate healing, and shorten the duration of viral shedding thereby reducing the incidence, severity and duration of pain, and limiting neuron damage [44]. The effect on the resolution of pain is extremely important in the antiviral therapy of HZ. Pain associated with HZ can be measured in three ways: (i) pain at presentation (acute pain), quantified over the first 30 days; (ii) PHN (post-herpetic neuralgia), defined as “pain that has not resolved after 30 days of disease onset" or as “pain that persists after healing or pain 90 days after rash onset” and (iii) zoster-associated pain (ZAP), pain recorded from the time of acute zoster until its complete resolution [44]. 

Acyclovir, valacyclovir and famciclovir are approved worldwide for the treatment of HZ in both immunocompetent and immunocompromised patients, brivudine is available in some European countries, and amenamevir is licensed only in Japan. Valacyclovir proved superior to acyclovir according to ZAP analysis from different clinical studies [44]. Famciclovir and acyclovir were therapeutically equivalent in terms of healing rate and pain resolution in immunocompetent patients aged >50 years. Brivudine proved similar efficacy on pain and rash as well as a similar tolerability compared to famciclovir in a large multicenter study that enrolled patients with acute HZ aged ≥50 years [44]. 

However, existing antiviral therapies are not completely effective in avoiding PHN, most likely because antivirals should be started within 72 h of rash appearance. The delay between onset of symptoms and start of treatment is likely the major cause of reduced efficacy of antiviral therapy. Clinical trials of antiviral drugs for HZ have enrolled patients within 72 h from rash onset; however, no well-controlled clinical trials comparing early-onset therapy with later therapy (>72 h) have been performed. Therefore, antiviral agents with a higher potency may achieve a more rapid decrease in viral replication consequently reducing neural damage and both acute and chronic symptoms of HZ. In addition, currently available antivirals require 3–5 times daily dosing regimens that need to be modified for patients with renal impairment. The medications used to treat the pain associated with PHN are only palliative, are often insufficient in terms of relief for the patients and do not provide a cure for HZ. Hence, drugs with superior anti-VZV activity, with the ability to prevent PHN, able to provide better pain relief, and with a more simplified dosing regimen are indeed needed. These drugs would also be very useful for the management of disseminated VZV primary infections and complications of VZV reactivation, including VZV vasculopathy, meningoradiculitis, cerebellitis, myelopathy, ocular disease, and zoster sine herpete (ZSH, radicular pain in the absence of skin rash).

### 5.2. Managing Rare but Significant Side Effects Linked to the VZV Life Vaccine

The varicella vaccine in infants diminishes the consequences of chickenpox in terms of both healthcare and economic burden, while the zoster vaccine protects immunocompetent adults from HZ and reduces disease severity in those who develop HZ. Adverse events were not seen in clinical trials performed with the VZV vaccines. However, rare but important side effects are being increasingly described most likely due to the increased administration of VZV life vaccines worldwide. 

A few cases of disseminated varicella infections due to the VZV vaccine strain were described in immunocompromised patients, requiring treatment with antiviral agents [45,46]. VZV vaccine can occasionally reactivate in healthy children and cause HZ [47,48] and in adults [49]. A disseminated VZV infection with CNS involvement directly following vaccine administration has been reported in a previously healthy elderly woman [50]. Herpes zoster and meningitis due to reactivation of the VZV vaccine virus has been described in an immunocompetent child [51]. Also, a case of vaccine-associated HZ ophtalmicus and encephalitis in an immunocompetent child requiring acyclovir therapy has been described [52]. 

Clinicians should also be aware that very rarely, the varicella vaccine can be transmitted and cause invasive disease as shown by a case report showing that the VZV vaccine was transmitted within a family from a child with shingles resulting in varicella meningitis in an immunocompetent adult [53].

Therefore, antiviral therapy will be needed for treatment of rare VZV diseases even after widespread implementation of vaccination programs.

### 5.3. Emergence of Drug-Resistant Viruses

Treatment of VZV infections with acyclovir or valacyclovir in the immunocompetent hosts is not associated with emergence of. In the immunocompromised hosts, VZV infection tends to be severe and persistent, requiring prolonged therapy with antivirals and ACV-R mutants have been isolated after long-term treatment with acyclovir [54,55,56,57,58]. Persistent VZV and VZV-related complications occur more frequently among hematological patients and antiviral resistance was found in 27% of patients with persistent VZV, including patients that progressed to severe retinal or cerebral infection [59]. Besides, compartmentalization of ACV-R VZV has been reported with important implications for sampling in molecular diagnostics [60]. Notably, ACV-R VZV keratitis was reported in an immunocompetent patient [61], highlighting the need to determine the antiviral-resistance patterns of corneal VZV isolates from chronic keratitis even in immunocompetent patients, as the eye can be considered an immune privileged site.

Resistance to acyclovir in VZV appears because of mutations in either the TK or the DNA polymerase genes. The most frequent mutants isolated both in cell culture and in the clinic are TK mutants [62,63]. Mutations associated with resistance to nucleoside analogues are distributed throughout the entire VZV TK gene, similarly to HSV-1 and HSV-2. However, conserved regions, such as the ATP- and the nucleoside-binding sites, and the amino acid position 231 are considered as hot spots for drug-resistance mutations [56,57,64,65,66]. Several case reports documented the use of foscarnet as salvage therapy for ACV-R VZV infections in immunocompromised patients [67,68,69]. However, mutations in the VZV DNA polymerase gene associated with PFA-R have also been described in immunocompromised patients [70,71,72]. The amino acid changes in the VZV DNA polymerase linked to PFA-R show generally cross-resistance to acyclovir. Remarkably, acyclovir and penciclovir were shown to select in vitro for different types of drug-resistant VZV genotypes: TK mutants under acyclovir pressure and DNA polymerase mutants under penciclovir selection [63,73]. This is different from HSV findings, where both drugs selected in vitro for TK mutants. Penciclovir remains active against some HSV-1 and VZV TK and DNA polymerase mutants that show ACV-R [74,75,76], indicating that the interactions between human α-herpesviruses TK and penciclovir or acyclovir, and also between the viral DNA polymerases and PCV-TP or ACV-TP are dissimilar, which may explains the differences between ACV-R and PCV-R VZV strains. Additionally, the frequency of VZV mutants was significantly higher following acyclovir pressure than penciclovir exposure [77]. As cidofovir (CDV, HPMPC) (Table 2) is a nucleotide analogue that is converted to the active form CDVpp, by cellular kinases, it is a therapeutic option for therapy of ACV-R, PCV-R and/or PFA-R resistant VZV infections [21,78,79].

Emergence of ACV-R in the course of a chronic infection caused by the Oka vaccine VZV strain was reported in an immunosuppressed child vaccinated prior a tumor diagnosis requiring intensive antitumor therapy [80]. Clinical ACV-R in this patient was linked to a TK mutation that responded to foscarnet therapy. ACV-R in the Oka vaccine strain was also described in a child with neuroblastoma [69], suggesting that the Oka vaccine strain can be linked to severe disease in the immunocompromised host following reactivation from latency, being necessary prolonged acyclovir therapy with the subsequent risk of emergence of drug-resistance. Hence, novel potent anti-herpesvirus agents with a target other than the viral DNA polymerase would be very useful to manage drug-resistance to the currently available antiviral agents.

## 6. Novel Anti-VZV Agents in Advanced Development

The gold standard for VZV therapy remains acyclovir and its prodrug valacyclovir. Other nucleoside analogues such as penciclovir, its prodrug famciclovir and brivudine can also be used. These antiviral agents rely on the viral TK for their first phosphorylation and have as target the viral DNA polymerase. VZV mutants arising under pressure with these nucleoside analogues bearing mutations in the viral TK can be treated with foscarnet, a direct inhibitor of viral DNA polymerases. However, foscarnet can be associated with significant renal toxicity and cannot be used for VZV DNA polymerase mutants emerging under acyclovir as most of them show cross-resistance to foscarnet. As cidofovir is a nucleotide analogue that bypass the activation by the viral TK and usually DNA polymerase mutants that are resistant to acyclovir and/or foscarnet remain sensitive to cidofovir, this drug can be used for VZV infections refractory to acyclovir, penciclovir and/or foscarnet. Unfortunately, cidofovir can also cause renal toxicity. Amenamevir, an helicase-primase inhibitor, has only been approved in Japan and its development was discontinued in United States due to toxicity concerns. The drawbacks of the currently available treatments for VZV-associated diseases highpoint the necessity for new, safe and highly effective anti-VZV agents. Drugs able to inhibit virus growth by targeting different steps of the VZV replicative cycle will be very useful for the management of drug-resistance infections in the clinic as well as for limiting the probability of emergence of antiviral drug-resistance and could also form the base for combination therapy.

Research should focus on the discovery and development of new anti-herpesvirus compounds having more potent activity than the currently available VZV antivirals. Besides, the search of new lead compounds able to block viral enzymes other than the viral DNA polymerase should be favored. In the last decade, only a few anti-VZV drugs were compared to valacyclovir in clinical trials to evaluate their efficacy in diminishing HZ associated pain and severity. 

### 6.1. Bicyclic Nucleoside Analogues (BCNAs)

In 1999, the potent and selective anti-VZV activity of some unusual bicyclic nucleoside analogues (BCNAs) was reported by Mc Guigan and collaborators [81]. For BCNAs with simple alkyl side-chain on the bicyclic base ring, the optimal carbon chain length for anti-VZV activity ranked between 8 and 10, with the 6-octyl-substituted derivative Cf-1368 being the most active and selective compound of this series of BCNAs. 

Although BCNAs are structurally related to BVDU, they differ in their spectrum of antiviral activity as they exclusively inhibit VZV, in contrast to BVDU that possesses potent anti- HSV-1 and anti-VZV activity. Among BCNAs with an aromatic ring system (phenyl) in the side-chain, the n-pentylphenyl- and n-hexylphenyl-derivatives (Cf-1742 and Cf-1743, respectively) (Figure 3) emerged as the most potent compounds with 50% effective concentration (EC_50_) values as low as 0.0001–0.0005 µM against reference VZV strains as well as clinical isolates in vitro [82,83]. A strong correlation between the length of the *n*-alkyl and *n*-alkylaryl moiety of the BCNAs and antiviral activity was observed [84]. Cf-1743 was also able to reduce the replication and spread of VZV in organotypic epithelial raft cultures as measured by morphological changes induced by the virus and quantification of the viral DNA load [85]. Similar to acyclovir and brivudine, the BCNAs were inactive against TK-deficient (TK−) strains, pointing to a crucial role of the VZV-encoded TK in the activation (phosphorylation) of the BCNAs. VZV mutant strains selected in vitro under pressure of BCNAs showed mutations in the viral TK gene [63].

BCNAs also strikingly differ from brivudine in their mechanism of activation (Figure 4) [86,87]. BCNAs are recognized by VZV TK as a substrate, but not by HSV-1 TK, nor by cytosolic TK-1 or mitochondrial TK-2 in kinetic studies with purified enzymes [84]. VZV TK was demonstrated to phosphorylate the BCNAs not only to their corresponding 5′-mono- but also to their 5′-diphosphate derivatives due to the intrinsic dTMP kinase activity of the VZV TK. Thus, BCNAs are selectively phosphorylated to their 5′ diphosphates by the two successive enzyme activities of VZV TK (thymidine kinase and dTMP kinase). However, no clear-cut correlation between the antiviral potency of the compounds and their affinity for VZV TK could be demonstrated, pointing to a different structure/activity relationship of the eventual antiviral target of these compounds. In contrast to BVDU, human NDP kinase was unable to convert BCNAs to their triphosphate derivatives (BCNA-TP) [84], in agreement with studies reporting no traces of BCNA-TP in VZV infected cells. These data clearly indicate that the mechanism of action of BCNAs differs from that of BVDU and suggests that BCNAs exert their antiviral effects via their monophosphate or diphosphate derivatives, virtually excluding the DNA polymerase as the molecular target of the BCNAs. The exact molecular target of the BCNAs could not be identified yet and is currently under investigation. 

BCNAs have also different catabolic pathways than BVDU and BVaraU (Figure 4), providing BCNAs with significant advantages. Pyrimidine nucleoside analogues are susceptible to pyrimidine catabolic enzymes (such as uridine phosphorylase or thymidine phosphorylase and are hydrolyzed to their free base metabolites that lack antiviral activity. In contrast, BCNAs were not recognized as substrates by human thymidine phosphorylase and thereby they were not converted to their inactive free base derivatives [88]. Furthermore, the free bases of BCNAs were not inhibitory to human dihydropyrimidine dehydrogenase, the catabolic enzyme involved in the degradation of pyrimidines analogues. As mentioned above, the BVU free base is inhibitory to dihydropyrimidine dehydrogenase, an enzyme necessary for the catabolic inactivation of the chemotherapeutic pyrimidine analogue 5-fluorouracil.

The safety of the oral prodrug of Cf-1743, i.e., FV-100 (valnivudine hydrochloride), was evaluated in three randomized, double-blind, placebo-controlled clinical trials: (i) a single-ascending-dose study in 32 healthy subjects aged 18 to 55 years (100-, 200-, 400-, and 800-mg doses); (ii) a multiple-ascending-dose study including 48 subjects (18 to 55 years-old) that received 100 mg once daily (QD), 200 mg QD, 400 mg QD, 400 mg twice a day, and 800 mg QD for 7 days); (iii) a two-part study in subjects aged 65 years and older with a single 400-mg dose in 15 subjects and a 400-mg QD dosing regimen for 7 days in 12 subjects [89]. FV-100 was shown to be rapidly and extensively converted to CF-1743; CF-1743 renal excretion was very low; high-fat but not a low-fat meal reduced exposure to CF-1743; and the FV-100 pharmacokinetic profile was similar in elderly and younger subjects. All doses maintained mean drug plasma levels of the active form of FV-100 that exceeded the EC_50_ for approximately 24 h, supporting the potential for once-a-day dosing in phase II trials. FV-100 was well tolerated at all doses and no serious adverse events were reported.

The safety of FV-100 and its efficacy in reducing pain associated with acute HZ compared to valacyclovir was evaluated in a prospective, multicenter, parallel-group, double-blind, randomized study [90]. Patients were ≥50 years old, had a diagnosis of HZ within 72 h of the formation of lesions and presented HZ-associated pain. They were randomized to receive a 7-day course of either FV-100 200 mg 1× per day (*n* = 117), FV-100 400 mg 1× per day (*n* = 116), or valacyclovir 1000 mg 3× per day (*n* = 117). Burden of illness (BOI), incidence and duration of clinically significant pain (CSP), pain scores, incidence and severity of PHN, and times to full lesion crusting and to lesion healing were used to evaluate efficacy. Safety was evaluated based on adverse event (AE)/ severe adverse events (SAE) profiles, changes in laboratory and vital signs values, and results of electrocardiograms. The BOI scores for pain through 30 days were 114.5 (200 mg FV-100), 110.3 (400 mg FV-100), and 118.0 (valacyclovir). The incidences of PHN at 90 days were 17.8%, 12.4%, and 20.2%, respectively, for FV-100 200 mg, FV-100 400 mg, and VACV. Adverse event and SAE profiles were similar in the two FV-100 and the valacyclovir groups. Although this trail missed the primary end-point (no statistically significant difference among the treatment groups for BOI at 30 days), a difference between FV-100 400 mg and valacyclovir groups was found in the 90 days data (14% reduction). These results point to a potential effect of FV-100 on subacute and chronic pain and demonstrate the potential utility of FV-100 as an antiviral for the treatment of shingles, diminishing both pain burden of the acute episode and incidence and severity of PHN. Reduced incidence of PHN, the most clinically meaningful endpoint, and reduced use of additional pain medication (opioids) for the management of PHN in the FV-100 arm vs. valacyclovir arm was seen. Compared to the standard of care valacyclovir, FV-100 offers secondary benefits over valacyclovir because FV-100 is once daily dosing vs. valacyclovir three-times dosing.

### 6.2. Carbocyclic Nucleoside Analogues: H2G (Omaciclovir) and Its Prodrug (Valomaciclovir)

H2G, (*R*)-9-[4-hydroxy-2-(hydroxymethyl)butyl]guanine, omaciclovir (Figure 3), a carbocyclic nucleoside analogue, has potent activity against different herpesviruses, being especially active against VZV, HSV and EBV but lacking activity against HCMV [91]. The average EC_50_ for omacyclovir was 2.3 ± 0.8 µM, compared to 46.8 µM for acyclovir and 78.7 µM for penciclovir against 20 VZV clinical isolates, pointing to a superior activity of omaciclovir against this virus [92]. 

Omaciclovir has a mode of action similar to that of acyclovir, but with less selectivity as a substrate for TK and resistance to omaciclovir maps to the TK [93]. However, omaciclovir, in contrast to acyclovir, is not an obligate chain terminator, although incorporation of the triphosphate form (omaciclovir-TP) results in limited chain elongation. Omaciclovir-TP has a longer intracellular half-life than ACV-TP (providing dosing advantages over acyclovir) and is a potent inhibitor of VZV DNA polymerase though less active than ACV-TP. The anti-herpesvirus activity of omaciclovir can be markedly enhanced by the immunosuppressive agent mycophenolate mofetil [94]. 

Since omaciclovir oral bioavailability is of 17% in cynomolgus monkeys, similar to that of acyclovir, the diester prodrug valomaciclovir stearate, EPB-348 [l-valine, (3*R*)-3[2-amino-1,6-dihydro-6-oxo-purin-9-yl]methyl]-4[(1-oxooctadecyl)oxo]butylester] (Figure 3) was synthesized with an oral bioavailability of >70% in rats and monkeys, and >60% in humans [95]. Omaciclovir was licensed from the Swedish biotech company Medivir AB to Epiphany Biosciences for further development under the name EPB-348. A phase IIb trial enrolled 373 immunocompetent patients with acute HZ, randomized into three arms: 1 g 1× daily valomaciclovir stearate, 2 g once daily valomaciclovir stearate, and 1 g 3× per day valacyclovir [96]. Eighteen patients also received 3 g of valomaciclovir stearate once daily. The primary endpoint was non-inferiority in terms of time to complete crusting of the shingles rash for the 1× daily valomaciclovir stearate cohort compared to valacyclovir cohort. Once daily valomaciclovir stearate 2 g met the primary endpoint, being more convenient than 3× daily valacyclovir for the treatment of HZ and also equally safe. Valomaciclovir stearate proved non-inferior to VACV in the secondary endpoints (time to complete pain resolution, time to rash resolution and time to cessation of new lesion formation). Moreover, the highest valomaciclovir stearate dose (3 g 1× daily) demonstrated superiority to valacyclovir for the primary endpoint with no significant adverse event differences between valomaciclovir stearate and valacyclovir groups, being nausea the most common adverse event in all patient groups. 

Valomaciclovir stearate proved also effective against acute infectious mononucleosis due to primary EBV infection, for which there is no FDA-approved treatment. The findings of a randomized, placebo-controlled, double-blind trial of valomaciclovir stearate for infectious mononucleosis were reported in 2009 at a conference but data have not yet been published [97]. Omaciclovir produced a significant decrease in median EBV load in the oral compartment compared to placebo but no differences were found in clearance of EBV DNAemia, CD8:CD4 ratios, CD8 lymphocytosis, or CD8 responses to lytic and latent EBV tetramers in the valomaciclovir stearate vs. placebo group. 

On Epiphany Biosciences’ website, omaciclovir stearate is still listed as active but without information on further development.

### 6.3. Brincidofovir

Brincidofovir (CMX001) (Table 2) is an orally bioavailable lipid acyclic nucleoside phosphonate (ANP) with the same broad-spectrum activity against DNA viruses as its parent compound cidofovir.

Because of the limitations associated with the use of cidofovir, the hexadecyloxypropyl (HDP) ester of cidofovir (CMX001, brincidofovir) (Table 2) was synthesized. In alkoxyalkyl esters prodrugs, a natural fatty acid (lysophosphatidylcholine) molecule is used as carrier to facilitate drug absorption in the gastrointestinal tract [98,99]. Brincidofovir has improved uptake and absorption, an oral bioavailability in mice of 88–97% (compared to less than 5% for cidofovir), increased cell penetration (10–20 fold) and higher intracellular levels (100-fold) of CDV-DP than those reached with cidofovir, resulting in superior antiviral activity. Furthermore, brincidofovir is not nephrotoxic because, in contrast to cidofovir, it is not a substrate of the human organic anion transporter 1 enzyme present in the proximal renal tubular cells. Despite promising preclinical data with brincidofovir, the results of a phase III trial evaluating its efficacy in the prevention of HCMV disease in seropositive allogeneic hematopoietic stem cell transplant patients were disappointing (SUPPRESS trial, NCT01769170) [100]. The end-point of the study was prevention of clinical significant HCMV infections 24 weeks’ post-transplantation. Unfortunately, a higher viral infection rate was found in the brincidofovir arm than in the placebo group (22% vs. 11%) as well as a higher proportion of patients developing graft-versus-host-disease (GvHD) and digestive symptoms. It is unclear if the emergence of digestive symptoms was linked to drug toxicity or if the drug favored the onset of digestive GvHD, for which the patients had to start immunosuppression explaining the increase in clinically significant HCMV infections. Based on the results of this study, Chimerix suspended further trials with oral brincidofovir for HCMV. In September 2019, SymBio Pharmaceuticals Limited (SymBio, Tokyo, Japan) announced an exclusive global license agreement with Chimerix Inc. for brincidofovir. Chimerix granted SymBio exclusive worldwide rights to develop, manufacture, and commercialize BCV in all human indications, except for the prevention and treatment of smallpox.

Regarding the clinical data of brincidofovir against VZV, a retrospective study including 30 HSCT recipients, provided limited-evidence on the potential efficacy of brincidofovir for prophylaxis of HSV and VZV in this group of patients [101]. The successful use of brincidofovir in management of disseminated acyclovir- and cidofovir-resistant VZV in a hematopoietic stem cell transplant patient with chronic GvHD who was intolerant to foscarnet has also been reported [102].

## 7. Candidate anti-VZV Drugs

### 7.1. Nucleoside/Nucleotide Analogues

#### 7.1.1. Phenoxazine Derivatives

Phenoxazine [1,3-diaza-2-oxophenoxazine] binds strongly to guanine in a duplex and improves TT-TT stacking interactions with adjacent bases. The phenoxazine scaffold is commonly employed to stabilize nucleic acid duplexes and fluorescent probes incorporating a phenoxazine component have been used for the study of nucleic acid structure, recognition, and metabolism. Phenoxazines are well-known in the field of medicinal chemistry because they have various biological activities, i.e., anticancer, antiviral, antidiabetic, antioxidant, anti-Alzheimer, anti-inflammatory, and antibiotic properties [103].

The antiviral activity against a panel of diverse viruses of newly synthesized phenoxazine-based nucleoside derivatives has been recently reported [104]. 3-(2′-Deoxy-β-d-ribofuranosyl)-1,3-diaza-2-ox-ophenoxazine (compound **7a**) (Table 3) proved to be a potent inhibitor of VZV replication with superior activity against wild-type than TK− strains (EC_50_ = 0.06 µM and 10 μM, respectively) [104]. Interestingly, compound **7a** was not cytotoxic or cytostatic for HEL cells at 100 μM (the maximum concentration tested), resulting in selectivity indices (ratio CC_50_/EC_50_) of 1667 (reference Oka strain) and 10 (TK− 07-1). The mechanism of action of phenoxazines derivatives against VZV is unknown but previously described phenoxazine derivatives with antiviral activity against HCMV, HSV-1 and HSV-2, were shown to directly inactive herpesviruses [105].

#### 7.1.2. 2′-Deoxyribose Emimycin Nucleosides

Emimycin (1,2-dihydro-2-oxopyrazine 4-oxide) is a pyrazine analogue structurally resembling uracil with known antibacterial properties. Emimycin, its 5-substituted congeners and the ribonucleoside derivatives are devoid of antiviral activity against RNA viruses. Interestingly, some of the 2′-deoxyribosyl emimycin derivatives proved potent inhibitors of the replication of HSV and VZV but were completely devoid of activity against HCMV [106]. The 2′-deoxyribonucleosides with an emimycin nucleobase (structurally similar to uracil) did not display inhibitory activity against VZV, HSV-1 and HSV-2. In contrast, the presence of 5-methylemimycin (related to the natural thymidine nucleobase) in compound **27a** (Table 3) conferred potent antiviral activity against VZV TK+ (EC_50_ = 0.99 μM) and HSV-1 TK+ (EC_50_ = 1.79 μM), with selectivity indices (ratio CC_50_/EC_50_) of 416 and 230, respectively. Compound **27a** was 7-fold (VZV) and 30-fold (HSV-1) less active when tested against mutant TK–strains and was devoid of anti-HCMV activity. The 5-ethylemimycin 2′-deoxyribose congener **28a** has a very similar profile as its 5-methyl congener **27a**, but its antiviral activity is less pronounced.

#### 7.1.3. C5-substituted-(1,3-diyne)-2-deoxyuridines

C5-substituted pyrimidine nucleosides, in particular those in the 2′-deoxyuridine series, constitute a unique class of compounds, playing an important role as components of nucleotide-derived tools for molecular genetics and as antiviral and anticancer agents. For instance, significant efforts have been expended to develop C5-substituted analogues that can be incorporated into DNA by viral DNA polymerases, such as the 5-(2-substituted vinyl)-2′-deoxyuridines, including brivudine. Novel C5-substituted-(1,3-diyne)-2′-deoxyuridines (with cyclopropyl, hydroxymethyl, methylcyclopentane, *p*-(substituted)phenyl and disubstituted-phenyl substituents) have been synthesized and evaluated against several herpesviruses [107]. The compound 5-[4-(4-trifluoromethoxyphenyl)buta-1,3-diynyl]-2-deoxyuridine (**26**, Table 3) emerged as the most potent inhibitor of this series against VZV with an EC_50_ of ∼1 μM and a CC_50_ of 55 μM. This compound had similar activity as acyclovir but it was less active than brivudine against the VZV TK+ YS and OKA strains and lost potency against TK− VZV strains (EC_50_ > 20µM). Compound **26** displayed moderate activity against HSV-1, HSV-2 and HSV-1 TK− with EC_50_ values of ~10–15 µM.

#### 7.1.4. Carbocyclic Nucleosides: C-3 halo and 3-methyl substituted 5′-nor-3-deazaaristeromycins

To expand on the antiviral properties of the carbocyclic nucleoside analogue 5′-noraristeromycin, 3-substituted 3-deaza-5′-noraristeromyin derivatives (i.e., bromo, iodo, chloro, and methyl) were synthesized [108]. An extensive characterization of their antiviral activities showed compound **4** (the bromo congener, Table 3) was most favorable towards the herpesviruses HCMV (EC_50_ = 1.7 µM), VZV (EC_50_ = 0.11 µM) without inhibiting cell growth at a concentration of 300 µM. This compound was also able to inhibit the replication of hepatitis B virus (HBV) and vaccinia virus but the activity against HSV was not reported.

#### 7.1.5. Xanthine-Based Acyclic Nucleoside Phosphonates (ANPs)

Noncanonic xanthine nucleotides XMP/dXMP play an important role in the balance and maintenance of intracellular purine nucleotide pool as well as in potential mutagenesis. A series of ANPs bearing a xanthine nucleobase were synthesized and evaluated for their activity against a wide range of DNA and RNA viruses [109]. Within this series, two ANPs, i.e., 9-[2-(phosphonomethoxy)ethyl]xanthine (PMEX) and 9-[3-hydroxy-2-(phosphonomethoxy)-propyl]xanthine (HPMPX), showed activity against several human herpesviruses. PMEX (Table 3), a xanthine analogue of adefovir (PMEA), emerged as the most important compound, exhibiting also activity against VZV (EC_50_ = 2.62 µM, TK+ Oka strain and EC_50_ = 4.58 µM, TK− 07-1 strain). The (S)HPMPX was less active with EC_50_ values of 22.7 µM and 17.1 µM, for respectively, TK+ and TK− VZV strains. The hexadecyloxypropyl monoester prodrug of PMEX (i.e., HDP-PMEX) proved 7- to 26-fold more active than the parent compound PMEX against VZV, although a concomitant increase in its cytostatic activity of 11-fold was also observed indicating successful increase in the cell uptake. The structures of HPMPX and HDP-PMEX are not reported here but can be found in the original study by Baszczynski and coworkers [109].

An increase in the EC_50_ of PMEX of the same magnitude as that measured for adefovir was found for well-characterized HSV-1 DNA polymerase mutant viruses indicating that the target of action of the active form of PMEX (i.e., PMEXpp) is the herpesvirus DNA polymerase. Furthermore, when the inhibitory activity of PMEXpp, was evaluated in an enzymatic assay against VZV DNA polymerase compared to cellular (α and β) DNA polymerases, PMEXpp proved inhibitory to VZV DNA polymerase when dGTP was used as competitive radiolabeled substrate with an IC_50_ = 7.4 µM without inhibition towards cellular DNA polymerases.

#### 7.1.6. Cyclopentyl Nucleoside Phosphonates

Both 2′-hydroxy-3′-deoxy- and 2′-deoxy-3′-hydroxycyclopentyl nucleoside phosphonates with the natural nucleobases adenine, thymine, cytosine and guanine were synthesized and evaluated for activity against several herpesviruses [110]. The guanine containing analogues displayed antiviral activity; in particular, the 3ʹ-deoxy congener **23** (Table 3) was active, with an EC_50_ of 5.35 μM against TK+ VZV strain and an EC_50_ of 8.83 μM against TK− VZV strain, while lacking cytotoxicity (CC_50_ > 289 µM). The application of phosphonodiamidate prodrug strategy did not result in a boost in antiviral activity.

#### 7.1.7. (*E*)-but-2-enyl Nucleoside Phosphonoamidates

A significant amount of research and development on aryl phosphoramidate prodrugs has been carried out by McGuigan’s group. However, the development of aryl phosphonoamidates, in particular in the field of ANPs, has been poorly investigated. Recently, the synthesis and antiviral evaluation of hitherto unknown (*E*)-but-2-enyl nucleosidephosphonoamidates using the cross-metathesis in water-under ultrasound irradiation has been reported by Agrofolio’s group [111]. The overall yield obtained was high, well above the previously data for the preparation of phosphonoamidates (15% vs. ~3%). Among the synthesized (*E*)-but-2-enylnucleoside phosphonoamidates, the thymine analogue **19** (Table 3) proved to be the best prodrug tested against VZV with an EC_50_’s of 0.39 and 0.33 µM for TK+ and TK− strains, respectively, and a selectivity index ≥200. This innovative synthetic approach may open the way for the synthesis of new purine and pyrimidine (*E*)-but-2-enyl phosphonoamidates.

#### 7.1.8. Prodrugs of C5-Substituted Pyrimidine Acyclic Nucleosides for Antiviral Therapy

While C5-Substituted-Uracil ANPs were devoid of antiviral activity, the bis(POM) prodrug of (*E*)-TbutP (compound **16**) (Table 3) presented a potent anti-herpesvirus activity, namely, HSV-1, HSV-2 and VZV (EC_50_ = 2.5−6.1 μM for these three viruses) [112]. Interestingly, the compound showed a decreased inhibitory effect (EC_50_ = 9.2 μM) against the TK− HSV-1 strain, but an increase in activity against the VZV TK− strain with EC_50_ = 0.19 μM compared to EC_50_ = 0.41 µM for VZV TK+ strain. There was no activity against HCMV. The (*E*)-bis(POM)-5Br-UbutP congener **18** proved weakly inhibitory against HSV-1, HSV-2, and VZV with EC_50_’s = 16−33 μM, and of 15–20 µM for VZV) but not against HCMV and kept activity against TK− HSV-1 and VZV strains. The compounds showed no cytotoxic at 200 μM, but were slightly cytostatic at ~35 μM for HEL cells. In contrast with the (*E*) isomer of bis(POM)-TbutP and bis(POM)-UbutP, the corresponding (*Z*) isomers exhibited negligible or no antiviral activity.

The phosphoro(no)amidate approach proved very successful both on nucleoside analogues (e.g., Sofosbuvir for hepatitis C virus) as well as on ANPs (e.g., Tenofovir alafenamide for HIV infection and HBV). Excellent results using the phosphoroamidate approach on adefovir were also obtained. In search for more potent and safe ANPs, the impact of the phosphonoamidate and phosphonodiamidate prodrugs on the antiviral activity of C5-pyrimidine acyclic nucleosides derivatives functionalized with but-2-enyl- chain was studied [113]. In the phosphonoamidate series, the most active compound **15** showed sub-micromolar activity against VZV (EC_50_ = 0.09–0.5 μM) and μM activity against HCMV and HSV. Notably, the phosphonodiamidate **21** showed excellent activity against wild type and TK− VZV strains (EC_50_ = 0.47 and 0.2 μM, respectively) and HCMV (EC_50_ = 3.5–7.2 μM) without cytostatic effect at the highest tested concentration (CC_50_ > 100).

#### 7.1.9. Prodrugs of the Pyrimidine Acyclic Nucleoside Phosphonates PMEO-DAPy and PME-5-azaC

Novel prodrugs of 2,4-diamino-6-[2-(phosphonomethoxy)ethoxy]pyrimidine (PMEO-DAPy) and 1-[2-(phosphonomethoxy)ethyl]-5-azacytosine (PME-5-azaC) prodrugs were synthesized with a pro-moiety consisting of carbonyloxymethyl esters (POM, POC), alkoxyalkyl esters, amino acid phosphoramidates and/or tyrosine (Table 3) [114]. PMEO-DAPy belongs to the class of open-ring ANPs, which are characterized by the phosphonomethoxy group containing an aliphatic part linked to the position 6 of 2,4-diaminopyrimidine via the oxygen atom. They are mimics of the appropriate 2,6-diaminopurine derivatives with an open imidazole ring. Interestingly, PMEO-DAPy has activity against retroviruses, HBV as well as herpesviruses. The replication of herpesvirus was inhibited by the bis(POC) and the bis(amino acid) phosphoramidate prodrugs of PMEO-DAPy with lower EC_50_ values than those for the parent compound PMEO-DAPy though the selectivity against HSV and VZV was only slightly improved. Thus, the VZV EC_50_ values and CC_50_ values for PMEO-DAPy were respectively, of 2.07–4.72 µM and 90.3 µM vs. ~0.3 µM and 15.1 µM [Bis(POC), compound **4**] and 0.56–1.69 µM and 25.9 µM [bis(amino acid) phosphoramidate, compound **5**]. The PME-5-azaC mono-octadecyl ester (compound **17**) was the most effective and selective PME-5-azaC prodrug when evaluated against HSV (EC_50_ of 0.15–0.35 µM), VZV (EC_50_’s of 0.32–0.79 µM) and HCMV (EC_50_’s of 0.24–1.12 µM) compared to a minimal anti-herpesvirus activity of the parent compound PME-5-azaC. Although the bis(hexadecylamido-L-tyrosyl) and the bis(POM) esters of PME-5-azaC proved to be very potent anti-herpesvirus compounds, their selectivity indices were reduced relative to the mono-octadecyl ester prodrug.

#### 7.1.10. Diamyl Aspartate Amidate Prodrugs of 3-Fluoro-2-(phosphonomethoxy)propyl Acyclic Nucleoside Phosphonates

Acyclic nucleosides bearing a 3-fluoro-2-(phosphonomethoxy)propyl (FPMP) side chain have moderate anti-HIV efficacy but are devoid of activity against DNA viruses. Two enantiomeric series of FPMP nucleosides bearing the four natural nucleobases was recently synthesized [115] (Table 3). Conversion of the phosphonic acid functionality of FPMPs into a diamyl aspartate phenoxyamidate group resulted in a novel generation of compounds having considerably improved antiretroviral potency as well as a larger spectrum of antiviral activity including HBV and herpesviruses. The (*S*)-FPMPA amidate prodrug displayed interesting activity against HIV (in the low nanomolar range), HBV and VZV. The (*S*)-FPMPA amidate prodrug (compound **22a**), one of the most active compounds, had 2000-fold better activity than the parent phosphonate against both TK+ and TK− VZV strains (EC_50_s in the 0.05 μM range). Compound **22a** was also active against HCMV but lacked selectivity for HCMV (CC_50_ = 1.96 μM for HEL cells, which was lower than the HCMV EC_50_ values) though it proved selective for VZV. Compounds **25a** and **25b** (both enantiomers of the guanine containing amidate prodrugs) showed similar activity and selectivity against VZV (EC_50_’s in the 0.079−0.89 μM range) and HCMV (EC_50_’s in the 1.98−5.94 μM range), but compound **25b** proved more active against HIV and HBV than its diasteromer **25a**. Among the pyrimidine analogues, compound **23a** (the cytosine containing (*S*)-Asp-prodrug) was the only congener displaying moderate activity against both VZV TK+ (EC_50_ = 3.15 µM) and TK− (EC_50_ = 1.32 µM) viruses and against HCMV [EC_50_ = 0.76 µM (AD-169 strain) and EC_50_ = 2.02 μM (Davis strain)].

#### 7.1.11. Amidate Prodrugs of Cyclic 9-(*S*)-[3-Hydroxy-2-(phosphonomethoxy)propyl]adenine, cHPMPA

The use of aryloxy monoamidates, a promising prodrug strategy for nucleoside phosphates and phosphonates developed the last years, was applied to HPMP-based compounds, resulting in the synthesis of phosphono amidate prodrugs of (*S*)-cHPMPA bearing different amino acid motifs [116]. All (*S*)-cHPMPA phosphonamidates prodrugs displayed broad-spectrum activity against herpesviruses with EC_50_ values in the low nanomolar range. Compound **8a** (cHPMPA diamyl aspartate phosphonoamidate prodrug) (Table 3) displayed good activity against HSV, VZV, and HCMV (EC_50_ values in the 0.0009−0.027 μM range, being the EC_50_’s for VZV of ~0.0005 µM) while the (*R*)-counterpart **8b** showed 24- to 180-fold lower potency. The (*S*)-Phe-cHPMPA (**10**), (*S*)-Glu-cHPMPA (**13**), (*S*)-Val-cHPMPA (**14**), (*S*)-Leu-cHPMPA (**15**), and (S)-Ile-cHPMPA (**16**) had also very potent anti-herpesvirus activity, with EC_50_ values of 0.00045−0.0043 µM for VZV. The structures of compounds **8b**, **10**, **13**, **14** and **15** are not shown here but were reported by Luo and colleagues [116]. Among these prodrugs, compound **16** (Table 3) inhibited VZV with EC_50_’s of 0.00045–0.00054 µM and emerged as the most selective one (selectivity indices’ of 20,000 and 1800, for VZV and HCMV, respectively). All prodrugs showed improved antiviral properties compared to the parent compound, which can be explained by their increased lipophilicity leading to enhanced cellular permeability. Additionally, HPMPA prodrugs displayed more potent activity than the reference anti-herpesvirus drugs and were equally active against wild-type and TK− mutants of HSV and VZV. Furthermore, the leucine ester prodrug of (*S*)-cHPMPA as well as the phosphonobisamidate valine ester prodrug of (*S*)-HPMPA were shown to be stable in human plasma.

### 7.2. Non-Nucleoside Analogues

#### 7.2.1. Pyrazolo[1,5-c]1,3,5-triazin-4-one Derivative

A derivative of pyrazolo[1,5-c]1,3,5-triazin-4-one, coded as **35B2** (2-[(2,6-dichlorophenyl)methylthio]-3*H*-pyrazolo[1,5-c]1,3,5-triazin-4-one) (Table 3) was identified from a library of 9600 random compounds as a novel anti-VZV compound screened by using a reporter cell line that produces luciferase upon infection with VZV [117]. The EC_50_ value for the vOka strain was 0.75 μM and the selective index was more than 200. The compound inhibited neither immediate-early gene expression nor viral DNA synthesis. Viral clones resistant to **35B2** had a mutation(s) in the amino acid sequence of the open reading frame 40 (ORF40), which encodes the major capsid protein, with most of the mutations located in the regions corresponding to the “floor” domain of the major capsid protein of HSV-1. Treatment with **35B2** altered the localization of major capsid protein in vOka-infected fibroblasts but not in fibroblasts infected with the drug-resistant clones. Normal major capsid protein localization in the **35B2**-treated infected cells was restored by overexpression of the scaffold proteins. Using electron microscopic analysis, lack of capsid formation could be demonstrated in the **35B2**-treated infected cells. These findings indicated a new class of anti-VZV agents targeting the herpesvirus major capsid proteins and inhibiting normal capsid formation.

#### 7.2.2. 5-Chlorobenzo[b]thiophen Derivative

From the screening of the 9600 compounds in the search of anti-VZV inhibitors, the compound [(5-chlorobenzo[b]thiophen-3-yl)methyl][(4-chlorophenyl)sulfonyl]amine, coded as **45B5** (Table 3), was identified also as a new anti-VZV agent [118]. Its EC_50_ against VZV vOka in HEL was 16.9 µM and its CC_50_ was >100 µM. The EC_50_ values against the parental Oka strain, acyclovir-resistant strains and **35B2**-resistant strains were similar to that against the vOka strain. Treatment of cells with **45B5** led to a weak but significant decrease of viral DNA synthesis and IE62 expression. Several **45B5**-resistant viral clones were isolated and characterized, having all of them at least one mutation in the ORF54 that encodes the portal protein. No effects on interaction between the portal and scaffold proteins were found, suggesting that **45B5** may inhibit nuclear delivery of viral DNA.

#### 7.2.3. Thienylcarboxamide Derivative

The screening of 9600 compounds also allowed the identification of a thienylcarboxamide derivative, coded as **133G4** [N-(4-chlorophenyl)-5-nitro-3-thienylcarboxamide] (Table 3), which was effective not only against VZV (EC_50_ = 19.3 µM) but also against HCMV [119]. The compound displayed a selectivity of 27.8 and >35.5 µM against VZV and HCMV, respectively. It was concluded that **133G4** was able to inhibit the activation of early gene promoters by HCMV IE2 and VZV IE62 considering that: (i) in HCMV-infected cells, **133G4** reduced HCMV early and late genes expression but not HCMV immediately early (IE1/IE2) gene expression, (ii) in HCMV-infected cells, **133G4** inhibited the activation of various HCMV early gene promoters of transiently-transfected plasmids, and (iii) in transient transfection assays, the compound reduced activation of HCMV (or VZV) early gene promoters by HCMV IE2 (or VZV IE62) in absence of other viral proteins. The inhibition of early gene activation was cell dependent as it was demonstrated in human and African green monkey cell lines but not in rodent cell lines, being the compound not effective against murine CMV. It was then hypothesized that **133G4** may target a cellular factor used commonly in activation of human herpesvirus promoters, however, the compound did not inhibit the recruitment of some cellular proteins that have been well-characterized as involved in VZV IE62-dependent viral gene activation to their binding sites, nor inhibited the direct interactions of VZV IE62 with cellular proteins.

#### 7.2.4. Indole-Based Derivatives

A library of indole-based derivatives was designed, synthesized and evaluated for antiviral activity against a broad spectrum of viruses [120]. The N-biphenylethyl acetamide derivative **17a** (Table 3) displayed significant inhibitory activity against VZV replication, including TK+ and TK− VZV strains, pointing to a mechanism of action independent from the virus-encoded thymidine kinase. Compound **17a** inhibited VZV plaque formation with EC_50_’s of 1.7–3.6 µM and a selectivity of 10–20 (ratio MCC/ EC_50_), being inactive against a variety of RNA and DNA viruses. A structure-activity relationship study showed that the substitution on the tryptamine moiety strongly influenced the compounds’ activity/toxicity. The concomitant presence of a biphenyl and methyl(phenyl) carboxy group at the amino group of tryptamine was required for anti-VZV activity.

#### 7.2.5. Cephalotaxine Esters

The alkaloid esters isolated from the genus *Cephalotaxus*, harringtonine (HT) and homoharringtonine (HHT) (Table 3), exhibit antitumor activity. Indeed, a semisynthetic HHT, omacetaxine mepesuccinate, has been approved by the FDA for treatment of chronic myelogenous leukemia patients resistant or intolerant to tyrosine kinase inhibitors. In addition to their known antitumor activity, HT and HHT possess potent antiviral activity with HT inhibiting chikungunya virus replication through down-regulation of viral protein expression and HHT displaying activity against HBV and coronavirus. HT and HHT, but not their biologically inactive parental alkaloid cephalotaxine, significantly hampered VZV replication of the recombinant VZV-pOka luciferase strain and a clinical isolate of VZV in vitro though the anti-VZV activity was lower for the clinical isolate than the reference strain [121]. HT and HHT reduced plaque formation with an estimated EC_50_ of 16.15 and 9.96 ng/mL, respectively. Their selectivity indices were low considering their CC_50_ values, i.e., 45.82 ng/mL (HT) and 74.71 ng/mL (HTT). The inhibition of VZV replication by HT and HHT was related to down-regulation of VZV lytic genes.

Given that HT and HHT inhibit replication of both RNA and DNA viruses, antiviral activities could be due to effects on cellular rather than viral factor(s) required for viral replication. HT and HHT were suggested to interfere with signal transduction pathways and transcription factors involved in VZV lytic gene expression. Furthermore, HT and HHT were reported to block cell cycle progression at G1/S or G2/M transition and considering that cell cycle regulation affects herpesvirus lytic gene expression and replication, a plausible mechanism to explain the anti-VZV activity of HT and HHT may be through effects on cell cycle in infected cells [120]. This study supports the potential of HT and HHT as candidates for treatment of VZV-associated diseases.

### 7.3. Hybrid Molecules

#### Dihydropyrimidinone/1,2,3-triazole Hybrid Molecules

A feasible strategy for producing drug molecules with potent activity is to combine two bioactive molecules belonging to different therapeutic categories. For instance, 1,2,3-triazole-based heterocycles have been used for the generation of many medicinal scaffolds with inhibitory properties against several viruses, including VZV. The multi-functionalized pyrimidinone scaffold represents a class of heterocyclic compounds with known pharmacological efficiency, including antiviral activity. Therefore, the synthesis of novel hybrid molecules can be regarded as a way to increase activity and/or decreased toxicity of the molecules. Novel hybrid compounds have been synthesized by combining the structural features of dihydropyrimidinone and 1,2,3-triazole heterocycles. Some of the tested compounds showed valuable antiviral activities, with EC_50_ values ranging from 3.6 to 11.3 μM against TK+ and TK− VZV and without measurable cell-growth inhibition. Compound **8a** (Table 3) emerged as the most active derivative with antiviral activities against VZV at EC_50_ of 3.62 μM (TK+ strain) and 7.85 µM (TK− strain) and CC_50_ >100 µM, followed by compound **7d** [EC_50_ of ~10–11 µM and CC_50_ >100 µM] [122]. None of the tested compounds exhibited measurable antiviral activity against HCMV. These findings suggest a selectivity of dihydropyrimidinone/1,2,3-triazole hybrid molecules into the herpesviridae family against VZV.

## 8. Conclusions and Perspectives

Despite the availability of a vaccine for the prevention of pediatric varicella in children and for the prevention of HZ in adults, there will continue to be a need for antiviral drugs. Some people in the elderly category are not able to mount a strong response to the vaccine. In immunocompromised persons, including patients with advanced HIV infection, varicella vaccination should be done exclusively with the subunit HZ vaccine as the life-attenuated vaccine is contraindicated for fear of disseminated vaccine-induced disease. On the other hand, the vaccine coverage is still quite limited and childhood immunization with lower coverage could theoretically shift the epidemiology of the disease leading to an increase in the number of severe cases in older children, teenagers and adults.

Infections with VZV are a serious cause of morbidity and mortality among immunosuppressed patients and in the elderly as PHN (the major complication of HZ) can be very difficult to manage. Currently available antiviral agents can shorten the duration of HZ and can promote rash healing, though they are not quite effective in preventing PHN, most likely due to the delay between diagnosis and the start of antiviral treatment. New antiviral chemotherapeutics with a different mechanism of action are under development for the management of HZ, including valnivudine hydrochloride (FV-100), prodrug of the bicyclic nucleoside analogue Cf-1743, the helicase-primase inhibitor amenamevir (ASP2151) that got only approval in Japan because of toxicity issues, and the carbocyclic nucleoside analogue valomaciclovir stearate (EPB-348), the omaciclovir prodrug. Although the development of amenamevir has been halted due to problems of toxicity following a trial in United States, the results with valnivudine hydrochloride and valomaciclovir stearate indicated that these drugs are effective, well-tolerated, with once-daily therapy regimen in the treatment of HZ. Further studies are needed to prove an advantage of valnivudine hydrochloride (FV-100) and valomaciclovir stearate over the standard of care valacyclovir. The development of other helicase-primase inhibitors and new drugs with a different target against VZV would be of value to manage ACV-resistant strains. Furthermore, inhibition of multiple targets in the viral replication cycle has the potential for synergistic effects when used in combination therapy, which would be of particular importance for immunocompromised hosts because it may avoid the need for extended therapeutic regimens and the risk of developing antiviral resistance during prolonged monotherapy. A potential anti-VZV drug candidate needs to be at least as effective and safe as the gold standard of treatment for HZ, i.e., valacyclovir. It will need to have also some advantages over valacyclovir, such as longer intracellular half-life allowing once a day dosing, superior efficacy, independence of TK for activation and/or target another enzyme than the DNA polymerase.

Antiviral agents will still have a role in the treatment of VZV-associated diseases. A large percentage of the HZ at-risk population does not receive the vaccine. The currently FDA-approved antiviral drugs, including acyclovir, valacyclovir and famciclovir have low efficacy in the control of pain for HZ patients and a significant proportion of these patients (~20–40%) develops PHN. These antivirals require multiple dosing regimens daily that need to be modified for patients with renal failure. Treatment with currently approved antiviral agents is recommended for VZV-associated infections in immunocompromised individuals, where VZV infections can be very severe. Among immunocompetent patients, antiviral therapy is recommended for adolescents and adults suffering from varicella and for patients (especially those older than 50 years) with HZ. Antiviral therapy will also be valuable to treat rare but significant side effects caused by the VZV vaccine. Although development of drug-resistance in considered much less common in VZV than in HSV, emergence of VZV drug-resistance in an emerging concern among immunosuppressed patients who have received prolonged antiviral therapy. Novel antiviral chemotherapeutics with different mode of action than the currently available anti-VZV drugs are required for the treatment of ACV-resistant strains in the immunocompromised host. Furthermore, compounds with a better activity than the currently approved drugs will be extremely useful if they are able to shorten the time to complete crusting of the shingles rash, time to complete pain resolution and prevention of PHN.

In the medical literature, cases of coronavirus disease 2019 (COVID-19) and HZ co-infections are being reported. An atypical bilateral acute retinal necrosis was diagnosed in a COVID-19 positive 75-year-old woman, who was immunosuppressed because of chemotherapy for a relapse of diffuse large cell B-cell lymphoma (DLBCL) [123]. Tartari and colleagues reported four cases of HZ in COVID-19 patients over 65 years old, three of them being admitted to ICU, requiring mechanical ventilation, and developing a necrotic HZ on the second branch of the trigeminal nerve [124]. The fourth patient, under immune suppressive therapy because of a cardiac transplant, showed a more indolent COVID-19 disease not requiring hospitalization and developed a classical HZ. Shors reported on a 49-year-old who developed HZ lesions 7 days following the emergence of COVID-19 symptoms and despite receiving valacyclovir 1 g 3× daily within 12 h of the initial eruption, the patient developed severe herpetic neuralgia [125]. A case of an immunocompetent middle-aged man presented with COVID-19 and 2 days later developed HZ. In another report, two cases of HZ reactivation preceding the emergence of respiratory symptoms in COVID-19 patients were demonstrated [126]. Furthermore, HZ may occur in asymptomatic COVID-19 patients, as found in a 39-year-old immunocompetent man who proved COVID-19 positive but symptoms free and presented HZ with trigeminal neuropathy [127]. These studies suggest an association between COVID-19 and VZV reactivation [128]. Given the nature of VZV reactivation, most commonly linked to stress, advanced age, and impaired immune conditions, more co-infection cases are expected to follow, which will require not only management of COVID-19 but also of VZV disease. To date VZV reactivation in COVID-19 patients has been described mostly in immunocompetent individuals and it has been proposed that the acute COVID-19 disease, associated with physical and emotional stress, might be a trigger factor for the development of HZ. Covid-19-associated lymphopenia, occurring due to direct damage to the thymus and spleen or due to the induced cytokine storm may impair antiviral responses. Furthermore, corticosteroids are being used to manage COVID-19, which may also favor VZV reactivation. Therefore, reactivation of herpesviruses, in particular of VZV, could be an emerging complication of COVID-19. Considering the ongoing COVID-19 pandemic, the CDC recommends vaccination against shingles for individuals at high risk, in particular for patients ≥50 years old. Besides, more potent and selective anti-VZV agents would be important to treat the unusual presentations and complications of HZ in COVID-19 patients.

## Figures and Tables

**Figure 1 molecules-26-01132-f001:**
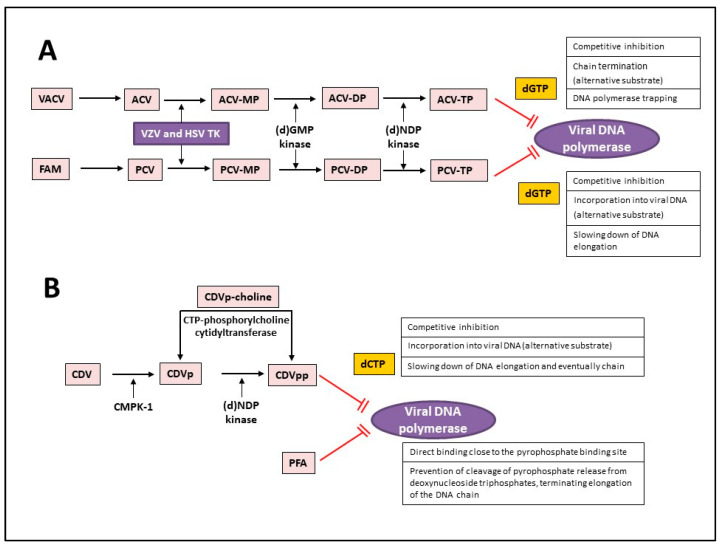
Activation and mechanism of action of (**A**) acyclovir (ACV) and penciclovir (PCV) and (**B**) cidofovir (CDV) and foscarnet (PFA). Valacyclovir (VACV), the oral prodrug of ACV, has improved absorption compared to ACV because of a stereo-selective transporter in the human intestine by dipeptide transporters followed by rapid and efficient hydrolysis to ACV by estereases found in the gut lumen, intestinal wall and liver. Famciclovir (FAM), the oral prodrug of PCV, follows first deacetylation at the 3 and 4 positions of the acyclic side chain and then oxidation at the 6 position of the purine ring yielding the active metabolite, PCV. The conversion to the monophosphate (MP) forms of ACV and PCV is carried out by the viral thymidine kinase (TK). The cellular enzyme guanosine monophosphate kinase or guanylate kinase (GMP) performs further phosphorylation to the diphosphate (DP) forms. Following conversion to their triphosphate (TP) forms by the cellular nucleoside 5′-diphosphate (NDP) kinase, the active metabolites inhibit viral DNA polymerases because they act as competitive inhibitors of the natural substrate (i.e., deoxyguanosine triphosphate, dGTP) and/or as alternative substrates when incorporated into the growing DNA chain. Incorporation of ACV-TP into the growing DNA chain results in chain termination due to the lack of an OH at the 5′ position. PCV-TP is not an obligate chain terminator due to the presence of an OH at the 5′ position and its incorporation results in slowdown of DNA chain elongation. The acyclic nucleotide analogue cidofovir (CDV) does not require activation by a virus-encoded enzyme for activation as the molecule already carries a phosphonate bond. In contrast to the O-P linkage (phosphate), the CH2-P-bond (phosphonate) is resistant to phosphodiesterase and phosphatase hydrolysis. Therefore, acyclic nucleoside phosphonates (ANPs), such as CDV, which mimic the nucleoside monophosphates, can bypass the initial enzymatic phosphorylation by viral kinases. Similar to a nucleoside monophosphate, a nucleoside phosphonate is further phosphorylated by cellular nucleotide kinases. The conversion of CDV to its active metabolite, i.e., CDV-diphosphate (CDVpp) is performed by cellular kinases [UMP/CMP kinase 1 (UMP/CMPK-1) and 5′-diphosphate (NDP) kinase]. CDV-DP, recognized by the viral DNA polymerase, will then block DNA synthesis by acting as competitive inhibitor with respect of the natural substrate dCTP or as alternative substrate leading to incorporation into the growing DNA. Chain termination occurs when two consecutive CDVpp’s are incorporated. CDVp-choline is regarded as an intracellular reservoir of CDVp and CDVpp. Foscarnet (PFA, phosphonoformic acid) does not require any activation by viral or cellular kinases and directly interacts with the viral DNA polymerase. PFA binds to the pyrophosphate exchange site of the viral DNA polymerase, blocking the release of pyrophosphate from the terminal nucleoside triphosphate and thus, impeding the formation of the 3′-5′-phosphodiester linkage essential for viral DNA elongation.

**Figure 2 molecules-26-01132-f002:**
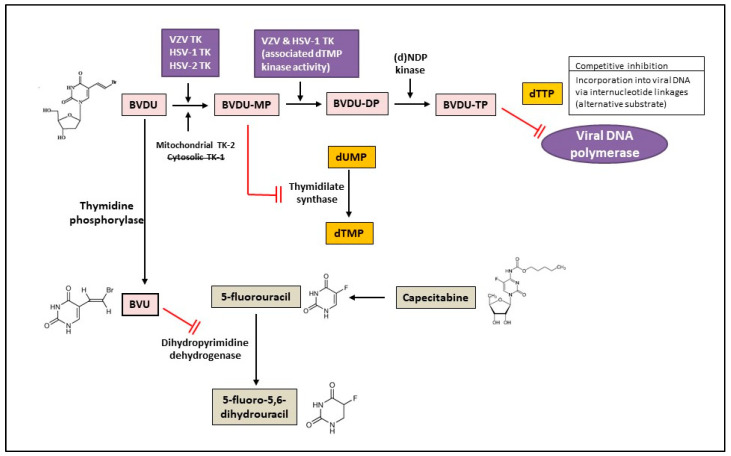
Activation, mechanism of action and catabolism of brivudine (BVDU). The VZV thymidine kinase (TK) as well as HSV-1 TK, display both thymidine kinase and thymidylate (dTMP) kinase activities, responsible for the activation of BVDU to the monophosphate (BVDU-MP) and diphosphate (BVDU-DP) forms, respectively. The conversion of BVDU-DP to the active triphosphate metabolite (BVDU-TP) is carried out by the cellular nucleoside 5′-diphosphate (NDP) kinase. BVDU-TP is recognized by DNA polymerases as an alternative substrate and is incorporated into the DNA molecule via internucleotide linkages. Pyrimidine nucleoside analogues, such as BVDU and BVaraU, can be degraded by pyrimidine catabolic enzymes (such as uridine phosphorylase or thymidine phosphorylase (TPase) leading to their free base metabolites without antiviral activity. BVDU cannot be administered together with 5-flurouracil or its prodrug capecitabine because BVU (the product formed following BVDU degradation by the thymidine phsophorylase), is a potent inhibitor of dihydropyrimidine dehydrogenase. This enzyme is needed for the first step in the catabolic pathway of pyrimidines and for 5-fluorouracil degradation and hence co-administration of 5-fluorouracil and brivudine leads to increased exposure to 5-fluorouracil.

**Figure 3 molecules-26-01132-f003:**
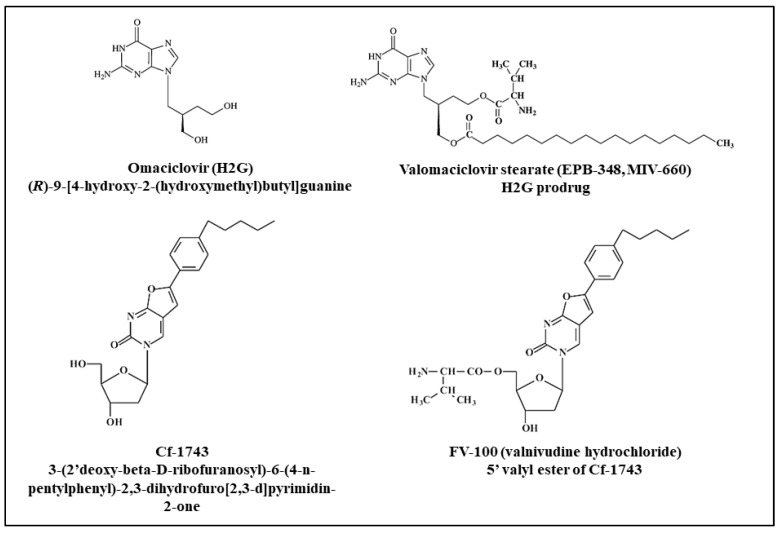
Anti-VZV drugs in advanced development. Chemical structures of CF-1743 and its prodrug valnivudine hydrochloride (FV-100) and of omaciclovir (H2G) and its prodrug valomaciclovir stearate.

**Figure 4 molecules-26-01132-f004:**
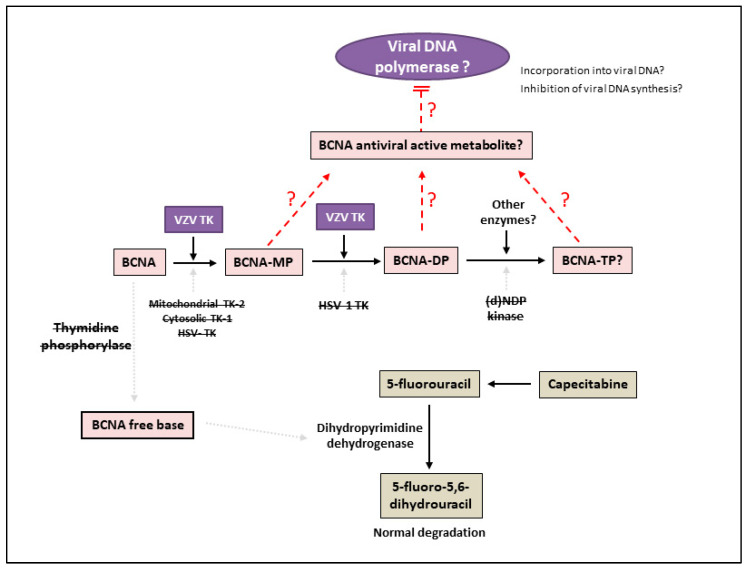
Activation, mechanism of action and catabolism of bicyclic nucleoside analogues (BCNAs). VZV TK converts BCNA’s to the mono- and diphosphate forms although whether there is conversion to the triphosphate form and which is the active metabolite and mechanism of action remain unclear to date. Striking differences between BCNAs and BVDU exist regarding their catabolic pathways. In contrast to BVDU, human TPases do not recognize BCNA’s as substrates and the free bases of BCNAs do not inhibit human DPD (dihydropyrimidine dehydrogenases) and thereof, there is a normal metabolism of Capecitabine/5-fluorouracyl. Dashed grey arrows indicate lack of activation.

**Table 1 molecules-26-01132-t001:** Vaccines available against varicella-zoster virus. Comparative characteristics of the available varicella zoster and herpes zoster vaccines.

	Varicella Vaccine	Herpes Zoster Vaccine	
Name	Varivax^®^ & Varilrix^®^	Zostavax^®^	Shingrix^®^
Year of FDA licensure	1995	2006	2018
Manufacturer	Merck (Varivax^®^) GSK (Varilrix^®^)	Merck	GSK
Type	Life-attenuated viral vaccine	Life-attenuated viral vaccine	Inactivated; Recombinant subunit
Vaccine components	Oka strain (1350 PFU)	Oka strain (19,400 PFU)	VZV glycoprotein E (gE)
Number of doses administered	Routine 2-dose vaccination 1st dose at 12–15 months old 2nd dose at 4–6 years old 2nd dose catch-up vaccination ≥3 months after 1st dose for children aged <13 years of age Adolescent & adults 2 doses 4 to 8 weeks apart	1 dose	2 doses (2–6 months apart)
Storage	Freezer	Freezer	Refrigerator
Diluent	Sterile water	Sterile water	Adjuvant
Dose form	0.5 mL vial for intramuscular injection	0.65 mL vial for subcutaneous injection	0.5 mL vial for intramuscular injection
Recommended age	≥ children 12 months Adults without evidence of immunity to varicella	FDA approved: ≥50 years old CDC recommendation: ≥60 years old	FDA approved: ≥50 years old CDC recommendation: ≥60 years old
Efficacy	About 98% protection in children and about 75% protection in teenagers and adults	Shingles prevention Age 50–59: 69% Age 60–69: 64% Age 70–79: 41% Age ≥80: 18% efficacy PHN protection Age 60–69: 51% Age 70–79: 64% Age ≥80: 41% Overall: 51%	Shingles prevention Age 50–59: 97.2% Age 60–69: 96.6% Age 70–79: 91.3% Age ≥80: 91.4% PHN protection Age ≥ 50: 91.2% Age ≥ 70: 88.8%
Immunity duration	Unknown; however, long-term efficacy studies have demonstrated continued protection up to 10 years after vaccination	Age-dependent	Age-dependent
Contraindications	Immunocompromised (primary or acquired) patients Family history of congenital immunodeficiency’s Individuals with anaphylactic reaction to any component of the vaccine efficacy	People with a history of severe allergic reaction (e.g., anaphylaxis) to any component of the vaccine Immunocompromised patients Individuals with family history of congenital immunodeficiency’s Pregnant and breastfeeding women	Individuals with a history of severe allergic reaction (e.g., anaphylaxis) to any component of the vaccine or after a previous dose of Shingrix. Pregnant and breastfeeding women Persons who currently have shingles
Possible side effects	Most commonly: Fever Pain, redness, and swelling at the injection site Varicella-like rash (transmission of varicella-virus can occur)	Most commonly: Pain, redness, and swelling at the injection site Muscle pain Tiredness Headache Shivering Fever Upset stomach	Most commonly: Pain, redness, swelling, warmth or itching at the site of injection at the injection site Headache

**Table 2 molecules-26-01132-t002:** Currently licensed anti-herpesvirus agents used to treat VZV infections. Anti-VZV drugs approved in United States, Europe and/or Japan for treatment of VZV-associated diseases and drugs used off label to treat acyclovir-resistant infections.

Drug	Prodrug	Mode of Action/Approval/Indication	Natural Analogue
**I. Nucleoside analogues**			
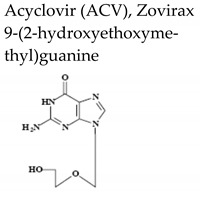	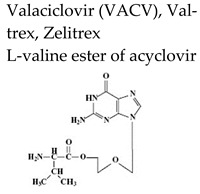	DNA Polymerase inhibitor—chain terminator at the incorporation site FDA approved Treatment and suppression of VZV infections (1st line treatment for VZV-associated diseases)	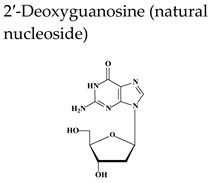
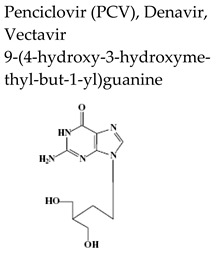	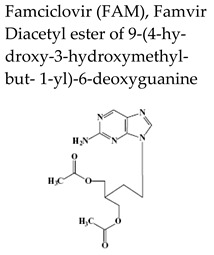	DNA Polymerase inhibitor—chain terminator after incorporation and elongation of several bases FDA approved Treatment of HZ	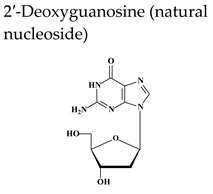
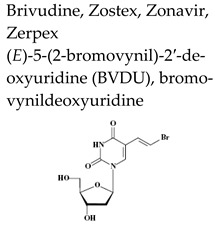	Orally bioavailable	Incorporation into viral DNA with replication-incompetent virus production Approved in some European countries Treatment of HZ	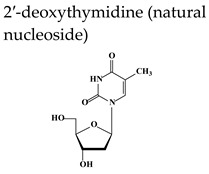
**II. Nucleotide analogue**			
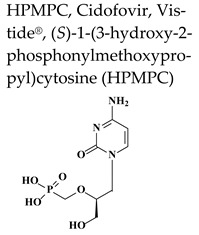	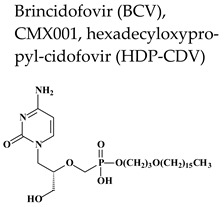	DNA Polymerase inhibitor—chain terminator after incorporation and elongation of several bases Cidofovir: off label use—treatment of ACV and/or PFA-resistant VZV infections Brincidofovir: herpesvirus development discontinued	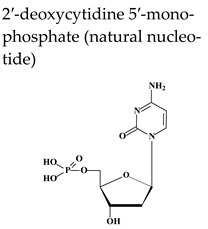
**III. Pyrophosphate analogue**			
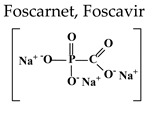		Direct DNA polymerase inhibitor without incorporation Off label use—treatment of acyclovir-resistant VZV infections	
**IV. Non-nucleoside analogue**			
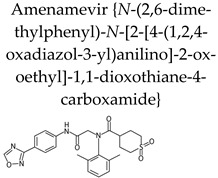	Orally bioavailable	Helicase-primase inhibitor Approved only in Japan Treatment of HZ	

**Table 3 molecules-26-01132-t003:** Candidate anti-VZV drugs. Compounds described in the last years showing promising activity against VZV.

Compound	EC_50_TK+ VZV	EC_50_TK− VZV	CC_50_	Activity against Other Human Herpesviruses
Nucleoside/nucleotide analogues
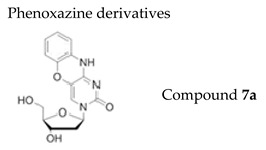	0.06 µM	10 µM	>100 µM	No activity against HCMV
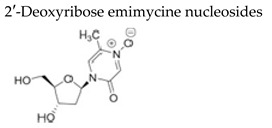	0.99 µM	6.91 µM	>412 µM	Activity against HSV-1, but lower activity against TK− HSV-1 and against HSV-2. No anti-HCMV activity
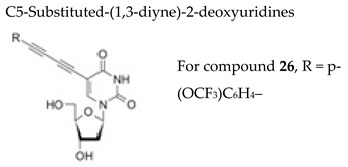	~1 µM	>20 µM	55 µM	Decreased activity against HSV-1 TK− and HSV-2.No anti-HCMV activity
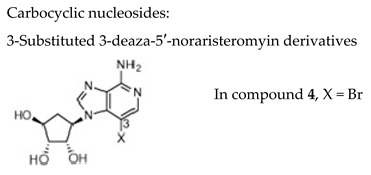	0.11 µM	No data available	>300 µM	Activity against HCMV, no data on other herpesviruses
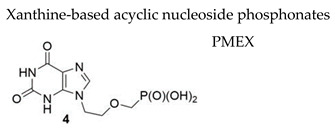	2.62 µM	4.58 µM	111 µM	Activity against HSV-1, HSV-2 and HCMV
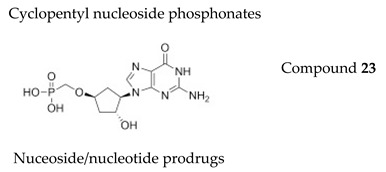	5.35 µM	8.83 µM	>289 µM	Activity against HSV-1, HSV-2 and HCMV
Nuceoside/nucleotide prodrugs
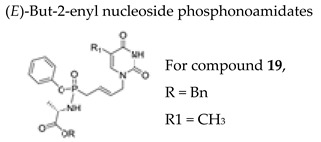	0.39 µM	0.33 µM	≥83 µM	Activity against HSV-1, HSV-2 and HCMV
Prodrugs of C5-substituted pyrimidine acyclic nucleosidesBis(POM) derivative of (*E*)-TbutP (compound **16**)Bis(POM) derivative of (*E*)-5Br-UbutP (compound **18**)				
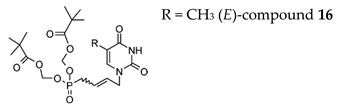	0.41 µM	0.19 µM	35 µM	Activity against HSV-1 & HSV-2 and no anti-HCMV activity
R = Br (*E*)- compound **18**	20 µM	15 µM	35 µM	Weak activity against HSV-1, HSV-2, no anti-HCMV activity
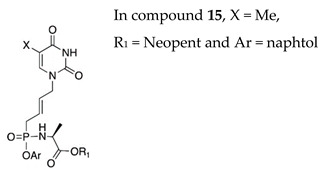	0.5 µM	0.09 µM	40 µM	Activity against HSV-1, HSV-2 and HCMV
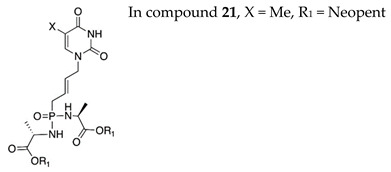	0.47 µM	0.20 µM	>100 µM	Activity against HSV-1, HSV-2 and HCMV
Prodrugs of pyrimidine acyclic nucleoside analogues				
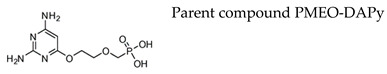	2.07 µM	4.72 µM	90.3 µM	Activity against HSV-1 & HSV-2 but not against HCMV
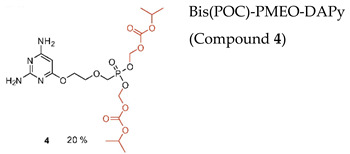	0.32 µM	0.29 µM	15.1 µM	Activity against HSV-1 & HSV-2 and marginal activity against HCMV
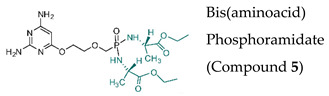	0.56 µM	1.69 µM	25.9 µM	Activity against HSV-1 & HSV-2 and low activity against HCMV
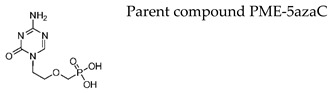	100 µM	40 µM	>200 µM	Marginal activity against HSV-1, HSV-2 & HCMV
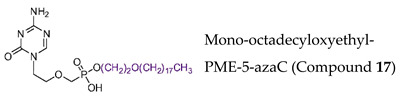	0.32 µM	0.79 µM	79.2 µM	Activity against HSV-1, HSV-2 and HCMV
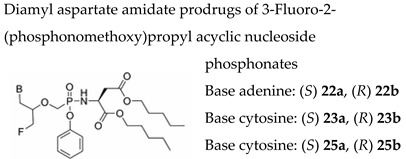				
**22a**	0.05 µM	0.057 µM	1.96 µM	No selectivity against HCMV
**23a**	3.15 µM	1.32 µM	74.7 µM	Activity and selectivity against HCMV
**25a**	0.59 µM	0.079 µM	27.8 µM	Activity but low selectivity against HCMV
**25b**	0.89 µM	0.11 µM	10.2 µM
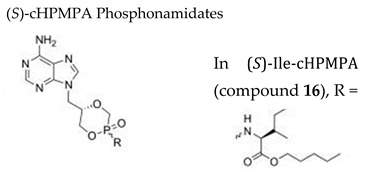	0.00045 µM	0.00054 µM	12 µM	Activity against HSV-1, HSV-2, and HCMV
Non-nucleoside analogues
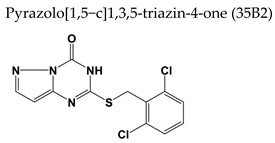	0.75 µM	4.6 µM	152.5 µM	Weak activity against HSV-1, no anti-HCMV activity
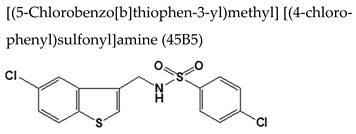	16.9 µM	18.3 µM	>100 µM	No data available on other viruses
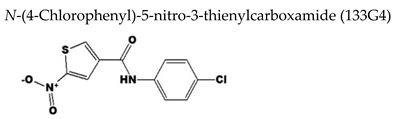	19.3 µM	No data available	537 µM	Activity against HCMV
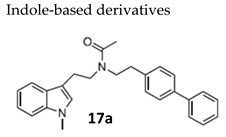	2.6 µM	2.0 µM	>100 µM	No activity against HSV-1, HSV-2, HCMV
Cephalotaxine esters				
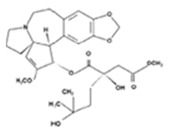	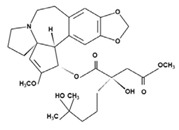	16.15 ng/mL (HT)9.96 ng/mL (HTT)	No data	45.82 ng/mL (HT)74.71 ng/mL (HTT)	No data available on other viruses—Low selectivity for VZV
HT	HTT				
Hybrid molecules
Dihydropyrimidinone/1,2,3-triazole hybrid molecules				
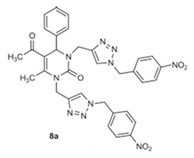	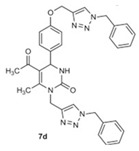	3.62 µM (**8a**)11.33 µM (**7d**)	7.85 µM (**8a**)10.79 µM (**7d**)	>100 µM (**8a**)>100 µM (**7d**)	No anti-HCMV activity

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
