# Peer review of "Advances and Perspectives in the Management of Varicella-Zoster Virus Infections"

_molecules, 2021, doi:10.3390/molecules26041132_

Round 1
Reviewer 1 Report
The review by Andrei and Snoeck is focused on varicella-zoster virus (VZV) diseases, introduction of vaccines to prevent VZV infection and description of anti-VZV agents both approved and described in literature. In particular, after describing the structure of VZV, they reported its mode of transmission, the ability to remain latent in the root ganglia and to eventually reactivate to give herpes zoster. Then, the primary clinical manifestation of VZV and the characteristics of HZ were well described, including the factor risks. Finally, the authors very clearly reported the numerous acute syndromes associated with VZV and HZ. The body of this review is represented by the description of vaccination strategies and current pharmacological treatments: analysis of all that concerns the management of infection complications, VZV life vaccine, drug-resistance, and description of the novel anti-VZV agents in clinical trials and promising candidates reported in literature.
This review is comprehensive, well organized and very clear. I only suggest a few small revisions.
- Pag 4 Table 1 Shingrix-Efficacy : “Shingles prevention”
- Line 179 “cannot excised”
- Lines 244-257 (caption of figure 1) the activation and mechanism of action of cidofovir is reported (its structure is also reported in table 1 among the approved drugs) but there is no description in the main text. It is described after in the paragraph “7.3 Brincidofovir”, which is the corresponding ester. This is a bit confusing. I would move the detailed description of cidofovir in paragraph 5, making a recall in paragraph 7.3.
- Lines 351-370 “Current antiviral drugs available for HZ treatment significantly decrease the incidence of new lesion formation, accelerate healing, and shorten the duration of viral shedding thereby reducing the incidence, severity and duration of pain [44]. The effect on the resolution… ……. is licensed only in Japan. All drugs decrease incidence of new lesion formation and accelerate healing and resolution of acute pain. Antiviral therapy shortens duration of viral shedding, and as a result limits neuron damage, decreasing incidence, severity and duration of pain. Valacyclovir proved superior to… ... that enrolled patients with acute HZ aged ≥50 years [44].” The same concept is reported in two paragraphs
- Line 378 semicolon to be deleted
- Line 398 “….and in”
- Lines 501-517 the concept was already reported (lines 212-219)
- Lines 537-538 (caption of figure 4) is referred to figure 2. In figure 4 BVDU is not reported
- Line 555, please add the references on clinical trials
- Lines 554-561 “The safety of the oral prodrug of Cf1723, i.e. FV-100 (Valnivudine hydrochloride), was evaluated in three randomized, double-blind, placebo-controlled clinical trials: (i) a single-ascending-dose study in 32 healthy subjects aged 18 to 55 years (100-, 200-, 400-, and 800-mg doses); (ii) a multiple-ascending-dose study including 48 subjects (18 to 55 years-old) that received 100 mg once daily (QD), 200 mg QD, 400 mg QD, 400 mg twice a day, and 800 mg QD for 7 days); (iii) a two-part study in subjects aged 65 years and older with a single 400-mg dose in 15 subjects and a 400-mg QD dosing regimen for 7 days in 12 subjects [89].”
- Line 573-578 “Burden of illness (BOI), 573 incidence and duration of clinically significant pain (CSP), pain scores, incidence and se-574 verity of PHN, and times to full lesion crusting and to lesion healing were used to evaluate 575 efficacy. and sSafety was evaluated based on adverse event (AE)/ severe adverse events 576 (SAE) profiles, changes in laboratory and vital signs values, and results of electrocardio-577 grams.”
- Line 697 various
- Line 720 “…in compound 27a (Table 3)…” in table 3 the number of compound (27a) is missing
- Table 3 compounds 27a, 26, 19, 16 (pag 20), 16 (pag 21), 35B2, 133G4: recheck the data, some of them do not correspond to those reported in the main text
- Line 759-768 “Within this series, two acyclic nucleoside phosphonates, …..” Please, specify that the structures of HPMPX and HDP-PMEX are not reported
- Check EC50 and CC50
- Line 873-878 “Compound 8a (cHPMPA diamyl aspartate phosphonoamidate prodrug) …...” Please, specify that the structures of 8b, 10, 13, 14 and 15 not reported
- Paragraph 8.2.5, there is no comment about the EC50 values
- Line 1069 COCVID-19
Author Response
Reviewer #1
Open Review
Comments and Suggestions for Authors
The review by Andrei and Snoeck is focused on varicella-zoster virus (VZV) diseases, introduction of vaccines to prevent VZV infection and description of anti-VZV agents both approved and described in literature. In particular, after describing the structure of VZV, they reported its mode of transmission, the ability to remain latent in the root ganglia and to eventually reactivate to give herpes zoster. Then, the primary clinical manifestation of VZV and the characteristics of HZ were well described, including the factor risks. Finally, the authors very clearly reported the numerous acute syndromes associated with VZV and HZ. The body of this review is represented by the description of vaccination strategies and current pharmacological treatments: analysis of all that concerns the management of infection complications, VZV life vaccine, drug-resistance, and description of the novel anti-VZV agents in clinical trials and promising candidates reported in literature.
This review is comprehensive, well organized and very clear. I only suggest a few small revisions.
We thank the Reviewer for his pertinent comments and we have revised the manuscript according to his/her suggestions
- Pag 4 Table 1 Shingrix-Efficacy : “Shingles prevention”
Changed.
- Line 179 “cannot excised”
Corrected.
- Lines 244-257 (caption of figure 1) the activation and mechanism of action of cidofovir is reported (its structure is also reported in table 1 among the approved drugs) but there is no description in the main text. It is described after in the paragraph “7.3 Brincidofovir”, which is the corresponding ester. This is a bit confusing. I would move the detailed description of cidofovir in paragraph 5, making a recall in paragraph 7.3.
The detailed description of cidofovir has been moved to session 5, as suggested by the Reviewer.
- Lines 351-370 “Current antiviral drugs available for HZ treatment significantly decrease the incidence of new lesion formation, accelerate healing, and shorten the duration of viral shedding thereby reducing the incidence, severity and duration of pain [44]. The effect on the resolution… ……. is licensed only in Japan. All drugs decrease incidence of new lesion formation and accelerate healing and resolution of acute pain. Antiviral therapy shortens duration of viral shedding, and as a result limits neuron damage, decreasing incidence, severity and duration of pain. Valacyclovir proved superior to… ... that enrolled patients with acute HZ aged ≥50 years [44].” The same concept is reported in two paragraphs
The second paragraph has been omitted.
- Line 378 semicolon to be deleted
Deleted.
- Line 398 “….andin”
Corrected.
- Lines 501-517 the concept was already reported (lines 212-219)
To avoid repetitions, lines 501-517 were removed.
- Lines 537-538 (caption of figure 4) is referred to figure 2. In figure 4 BVDU is not reported
Lines 537-538 (caption of figure 4) were moved to caption of figure 2, where BVDU is reported.
- Line 555, please add the references on clinical trials
References #89 is provided for the different clinical studies.
- Lines 554-561 “The safety of the oral prodrug of Cf1723, i.e. FV-100 (Valnivudine hydrochloride), was evaluated in three randomized, double-blind, placebo-controlled clinical trials:(i) a single-ascending-dose study in 32 healthy subjects aged 18 to 55 years (100-, 200-, 400-, and 800-mg doses); (ii) a multiple-ascending-dose study including 48 subjects (18 to 55 years-old) that received 100 mg once daily (QD), 200 mg QD, 400 mg QD, 400 mg twice a day, and 800 mg QD for 7 days); (iii) a two-part study in subjects aged 65 years and older with a single 400-mg dose in 15 subjects and a 400-mg QD dosing regimen for 7 days in 12 subjects [89].”
Paragraph edited.
- Line 573-578 “Burden of illness (BOI), 573 incidence and duration of clinically significant pain (CSP), pain scores, incidence and se-574 verity of PHN, and times to full lesion crusting and to lesion healing were used to evaluate 575 efficacy.and sSafety was evaluated based on adverse event (AE)/ severe adverse events 576 (SAE) profiles, changes in laboratory and vital signs values, and results of electrocardio-577 grams.”
Corrected.
- Line 697 various
Edited.
- Line 720 “…in compound 27a(Table 3)…” in table 3 the number of compound (27a) is missing
Compound 27 is now added to Table 3.
- Table 3 compounds 27a, 26, 19, 16 (pag 20), 16 (pag 21), 35B2, 133G4: recheck the data, some of them do not correspond to those reported in the main text
Values were checked and edited when necessary.
- Line 759-768 “Within this series, two acyclic nucleoside phosphonates, …..” Please, specify that the structures of HPMPX and HDP-PMEX are not reported
This is now specified in the text and a the reader is referred to a reference for the structures that are not provided.
- Check EC50and CC50
Checked.
- Line 873-878 “Compound 8a (cHPMPA diamyl aspartate phosphonoamidate prodrug) …...” Please, specify that the structures of 8b, 10, 13, 14 and 15 not reported
This is now mentioned in the text and the corresponding reference is provided referring to the structures.
- Paragraph 8.2.5, there is no comment about the EC50values
A comment is now included.
- Line 1069 COCVID-19
Edited.
Reviewer 2 Report
This manuscript reviews about the management of VZV infections. It is well reviewed and will provide useful information for readers. Therefore, I think this manuscript could be acceptable as a review in Molecules after considering some comments as below.
N, E, R, S, n (normal), and p in the compound names should be written in Italic style.
Compound No. should be written in bold style.
Figure 3 is indistinct. I suggest rewriting structures and the compound names to the authors.
Space should be inserted between the number and unit in all cases.
Where is compound 27a and 28a in Figure 3?
Author Response
This manuscript reviews about the management of VZV infections. It is well reviewed and will provide useful information for readers. Therefore, I think this manuscript could be acceptable as a review in Molecules after considering some comments as below.
We appreciate the comments of the Reviewer and we have modified tha manuscript according to his/her remarks.
N, E, R, S, n (normal), and p in the compound names should be written in Italic style.
Compounds’ names were checked and edited when necessary.
Compound No. should be written in bold style.
Done.
Figure 3 is indistinct. I suggest rewriting structures and the compound names to the authors.
A new figure 3 is now included with brincidofovir being removed as its structure is already provided in Table 2.
Space should be inserted between the number and unit in all cases.
Corrected.
Where is compound 27a and 28a in Figure 3?
The structures of the new compounds are provided in Table 3. It is now added compounds 27a.